# Batch Normalization for Neural Networks on Complex Domains

Xuan Son Nguyen [1]    Nistor Grozavu [1]

## Abstract

Riemannian neural networks have proven effective in solving a variety of machine learning tasks. The key to their success lies in the development of principled Riemannian analogs of fundamental building blocks in deep neural networks (DNNs). Among those, Riemannian batch normalization (BN) layers have shown to enhance training stability and improve accuracy. In this paper, we propose BN layers for neural networks on complex domains. The proposed layers have close connections with existing Riemannian BN layers. We derive essential components for practical implementations of BN layers on some complex domains which are less studied in previous works, e.g., the Siegel disk domain. We conduct experiments on radar clutter classification, node classification, and action recognition demonstrating the efficacy of our method.

## 1. Introduction

Neural networks on Riemannian manifolds have shown great success in a wide range of fields such as hierarchical and graph-structured data (Ganea et al., 2018; Chami et al., 2019), action and image classification (Huang et al., 2018; Nguyen et al., 2025b), brain-computer interface (Pan et al., 2022). While many frameworks for building such networks have been developed, most of them focus on well-known Riemannian manifolds, e.g., hyperbolic spaces (Ganea et al., 2018; Chami et al., 2019), Symmetric Positive Definite (SPD) spaces (Huang & Gool, 2017; Brooks et al., 2019; Nguyen et al., 2019; 2020; Nguyen, 2021; 2022a;b; Nguyen & Yang, 2023; Nguyen et al., 2024; 2025b), and Grassmann spaces (Huang et al., 2018; Nguyen & Yang, 2023). Although some frameworks were originally proposed in a wider context, e.g., matrix manifolds (Nguyen, 2022b; Nguyen & Yang, 2023; Chen et al., 2025) and Lie

groups (Chen et al., 2024), the assumptions based on which they are built limit their applicability.

Complex domains (open subsets of complex vector spaces) (Franzoni & Vesentini, 1980) have been encountered in a number of problems in signal processing (Barbaresco, 2013; Cabanes, 2022). Unfortunately, they have not received enough attention in the context of deep learning despite having many nice theoretical properties. For instance, Siegel spaces, which are special symmetric spaces of noncompact type (Helgason, 1979), generalize hyperbolic and SPD spaces. Recently, a few attempts (López et al., 2021; Nguyen et al., 2025a;b) have been made to explore the representation power of Siegel spaces in the context of deep learning. The work in (2021) was among the first to advocate the use of Siegel spaces for graph embeddings. However, it did not tackle the question of how to build fundamental building blocks for discriminative Siegel neural networks. Motivated by this, some building blocks were developed in (2025a; 2025b). Yet, none of these works extends BN layers to the geometry of Siegel spaces. Another example of complex domains is the unit ball in a complex Hilbert space (hereafter referred to as the complex unit ball). While the (real) Poincaré ball model of hyperbolic geometry has extensively been used and has demonstrated excellent performance in a variety of machine learning tasks (Ganea et al., 2018; Chami et al., 2019; Bdeir et al., 2024), the complex unit ball model is less investigated for these tasks.

Invariant metrics have shown to be important tools in complex analysis (Barbaresco, 2013). One of the most significant advances in the developments of such metrics was proposed in (1967) which generalizes the concept of the Poincaré distance (Helgason, 1979) to arbitrary complex domains. The resulting pseudodistance is referred to as the Kobayashi pseudodistance. An important property of the Kobayashi pseudodistance is that it is invariant for biholomorphic mappings (Kobayashi, 1967; Franzoni & Vesentini, 1980). This is a desirable property when it comes to building neural networks under the geometric deep learning framework in which the geometric stability principle plays a crucial role (Bronstein et al., 2017). Although this property makes the Kobayashi pseudodistance an attractive tool for the construction of neural networks in complex domains, it is less familiar in the context of deep learning compared to other distances (López et al., 2021; Nguyen et al., 2025a;b).

[1]ETIS, UMR 8051, CY Cergy Paris University, ENSEA, CNRS, Cergy, France. Correspondence to: Xuan Son Nguyen <xuanson.nguyen@ensea.fr>.

*Proceedings of the 43rd International Conference on Machine Learning*, Seoul, South Korea. PMLR 306, 2026. Copyright 2026 by the author(s).

In this paper, we are interested in building neural networks on complex domains. Our contributions are as follows:

- We propose a general BN layer for neural networks on complex domains. While such spaces are the focus of our work, this layer is also applicable to Riemannian manifolds widely used in machine learning tasks, and is closely related to existing Riemannian BN layers.

- We show how to apply the proposed BN layer to domains which are less studied in previous works despite their potential in capturing rich geometrical structures, i.e., the Siegel disk domain and the complex unit ball.

- We provide experimental evaluations showing that the proposed method yields systematic improvements with respect to existing methods.

## 2. Mathematical Background

In Section 2.1, we recap some concepts in complex spaces. We then present two complex domains in Sections 2.2 and 2.3. More discussions are provided in Appendix C. For greater mathematical detail and in-depth discussion, please refer to (1943; 1979; 1980; 2013; 2016).

### 2.1. Complex Domains and Invariant Distances

**Definition 2.1** (**Complex domains (Franzoni & Vesentini, 1980)**). Let $\mathbb{C}$ be the set of all complex numbers. Let $\mathbb{C}_n$ be the $n$-dimensional complex vector space along with its usual analytic structure. An open subset $\mathbb{D}$ of $\mathbb{C}_n$ is called a domain.

In the following, we use the terms "domains" and "complex domains" interchangeably. Similar to the case of real functions, one has the notion of differentiability for complex functions which is important in complex analysis.

**Definition 2.2** (**Holomorphic functions (Franzoni & Vesentini, 1980)**). Let $\mathbb{E}$ and $\mathbb{F}$ be complex normed spaces. Let $\mathbb{U}$ be a domain in $\mathbb{E}$. A mapping $f : \mathbb{U} \to \mathbb{F}$ is called a holomorphic function on $\mathbb{U}$ with values in $\mathbb{F}$, or an $\mathbb{F}$-valued holomorphic function if, for every $u \in \mathbb{U}$, there exists an open neighborhood $\mathbb{V}$ of $u$ in $\mathbb{U}$ such that $f(\cdot)$ can be expressed by

$$f(x) = \sum_{k=0}^{\infty} a_k (x - u)^k,$$

where $a_k \in \mathbb{C}, k = 0, \ldots, \infty$, and $x \in V \cap \mathbb{B}(u, r)$ (the open ball with center $u$ and radius $r > 0$). If a function $f : \mathbb{U} \to \mathbb{U}$ is holomorphic, one-to-one, and onto, and has a holomorphic inverse, then $f$ is referred to as an automorphism of $\mathbb{U}$.

On a complex domain, one can define a pseudodistance referred to as the Kobayashi pseudodistance (Kobayashi,

1967; Franzoni & Vesentini, 1980) which is invariant under automorphisms[1]. It is defined using holomorphic maps from the Poincaré disc $\mathbb{B}$ (see Appendix C.1) into the target domain. We first review the Kobayashi differential metric.

**Definition 2.3** (**Kobayashi differential metric (Franzoni & Vesentini, 1980)**). Let $\mathbb{E}$ be a complex normed space, and let $\mathrm{Hol}(\mathbb{D}_1, \mathbb{D}_2)$ be the set of all holomorphic functions $f : \mathbb{D}_1 \to \mathbb{D}_2$, where $\mathbb{D}_1$ and $\mathbb{D}_2$ are two domains in $\mathbb{E}$. Let $\mathbb{D}$ be a domain in $\mathbb{E}$. Let $x \in \mathbb{D}$ and $v \in \mathbb{E}$. The Kobayashi differential metric $g_{\mathbb{D}}^K : \mathbb{D} \times \mathbb{E} \to [0, +\infty)$ is defined as

$$g_{\mathbb{D}}^K(x, v) = \inf\{\frac{1}{\tau} : \tau > 0, h \in \mathrm{Hol}(\mathbb{B}, \mathbb{D}), h(0) = x,$$
$$h'(0) = \tau v\}.$$

The Kobayashi differential metric only gives a notion of infinitesimal length of a vector tangent to a curve joining two points in a domain. To compute the Kobayashi pseudodistance between two points, one needs to define the integrated form of a differential metric.

**Definition 2.4** (**Integrated form of a metric (Franzoni & Vesentini, 1980)**). Let $\mathbb{D}$ be a domain. The integrated form $\tilde{g}_{\mathbb{D}}(\cdot, \cdot)$ of a differential metric $g_{\mathbb{D}}(\cdot, \cdot)$ is defined as

$$\tilde{g}_{\mathbb{D}}(x, y) = \inf \int_a^b g_{\mathbb{D}}(\gamma(t), \gamma'(t)) dt, \tag{1}$$

where $x, y \in \mathbb{D}$, and the infimum is taken over all piecewise $C^1$ (continuously differentiable) curves $\gamma : [a, b] \to \mathbb{D}$ from $x$ to $y$, i.e., $\gamma(a) = x$ and $\gamma(b) = y$.

We can now define the Kobayashi pseudodistance.

**Definition 2.5** (**Kobayashi pseudodistance (Franzoni & Vesentini, 1980)**). The Kobayashi pseudodistance $d_{\mathbb{D}}^K(\cdot, \cdot)$ is the integrated form of the Kobayashi differential metric, i.e.,

$$d_{\mathbb{D}}^K(x, y) = \tilde{g}_{\mathbb{D}}^K(x, y),$$

where $x, y \in \mathbb{D}$, and the infimum in Eq. (1) is taken over all piecewise $C^1$ curves $\gamma : [0, 1] \to \mathbb{D}$ from $x$ to $y$, i.e. $\gamma(0) = x$ and $\gamma(1) = y$.

### 2.2. Siegel Spaces

In this section, we briefly review two models of Siegel spaces, i.e., the Siegel upper half space and the Siegel disk. We focus on the Siegel disk but a quick recap of the Siegel upper half space is essential for our exposition.

---

[1]We note that the Kobayashi pseudodistance can be defined for other settings, e.g., arbitrary complex manifolds. In this paper, we follow closely the work in (1980) which considers the setting of complex domains.

### 2.2.1. THE SIEGEL UPPER HALF SPACE

The Siegel upper half space $\mathbb{SH}_n$ is defined as

$$\mathbb{SH}_n = \{x = u + iv : u \in \mathrm{Sym}_n, v \in \mathrm{Sym}_n^+\},$$

where $\mathrm{Sym}_n$ and $\mathrm{Sym}_n^+$ denote the space of $n \times n$ real symmetric matrices and that of $n \times n$ SPD matrices, respectively.

### 2.2.2. THE SIEGEL DISK

The Siegel disk is an open convex complex matrix domain defined by

$$\begin{aligned}\mathbb{SD}_n &= \{x \in \mathrm{Sym}_{n,\mathbb{C}} : I_n - xx^H \in \mathbb{H}_n^+\} \\ &= \{x \in \mathrm{Sym}_{n,\mathbb{C}} : I_n - x^H x \in \mathbb{H}_n^+\},\end{aligned}$$

where $I_n$ is the $n \times n$ identity matrix, $x^H$ is the conjugate transpose of $x$, $\mathrm{Sym}_{n,\mathbb{C}}$ and $\mathbb{H}_n^+$ denote the space of $n \times n$ complex symmetric matrices and that of $n \times n$ Hermitian positive definite (HPD) matrices, respectively. In the following, the subscript $n$ is omitted from $I_n$ when it is clear from the context.

Let $\mathbb{U}_n$ be the set of $n \times n$ unitary matrices. Then any automorphism $\Psi(\cdot)$ of $\mathbb{SD}_n$ is written (Siegel, 1943) as

$$\Psi(y) = u\phi_x(y)u^T,$$

where $u \in \mathbb{U}_n$, $x, y \in \mathbb{SD}_n$, and the map $\phi_x(\cdot)$ is defined by

$$\phi_x(y) = (I - xx^H)^{-\frac{1}{2}}(y - x)(I - x^H y)^{-1}(I - x^H x)^{\frac{1}{2}}. \tag{2}$$

When $x$ varies on $\mathbb{SD}_n$, the set of maps $\phi_x(\cdot)$ is a subgroup acting transitively on $\mathbb{SD}_n$. The distance $d_{\mathbb{SD}_n}(\cdot, \cdot)$ is given (up to scaling) by

$$d_{\mathbb{SD}_n}^2(x, y) = \mathrm{Tr}\left(\log^2\left((I + c^{\frac{1}{2}})(I - c^{\frac{1}{2}})^{-1}\right)\right), \tag{3}$$

where $\mathrm{Tr}(\cdot)$ denotes the trace operator, $\log(.)$ denotes the matrix logarithm, and $c$ is computed as

$$c = (y - x)(I - x^H y)^{-1}(y^H - x^H)(I - xy^H)^{-1}. \tag{4}$$

### 2.2.3. CONVERSION BETWEEN THE TWO MODELS

One can convert a point $x \in \mathbb{SH}_n$ to $\mathbb{SD}_n$ using the following matrix Cayley transformation:

$$\varphi(x) = (x - iI)(x + iI)^{-1}.$$

The inverse matrix Cayley transformation that converts a point $x \in \mathbb{SD}_n$ to $\mathbb{SH}_n$ is given by

$$\varphi^{(-1)}(x) = i(I + x)(I - x)^{-1}.$$

### 2.3. The Complex Unit Ball

Let $x = (x_1, \cdots, x_n), y = (y_1, \cdots, y_n) \in \mathbb{C}_n$. We define the inner product $\langle \cdot, \cdot \rangle$ on $\mathbb{C}_n$ as

$$\langle x, y \rangle = \sum_{j=1}^{n} x_j \bar{y}_j,$$

where $\bar{y}_j$ is the complex conjugate of $y_j$, and we write

$$|x| = \sqrt{\langle x, x \rangle} = \sqrt{\sum_{j=1}^{n} |x_j|^2},$$

where the notation $|\cdot|$ on the right-hand side denotes the complex modulus. The complex unit ball is then defined as

$$\mathbb{B}_n = \{x \in \mathbb{C}_n : |x| < 1\}.$$

Any automorphism $\Psi(\cdot)$ of $\mathbb{B}_n$ is written (Franzoni & Vesentini, 1980) as

$$\Psi(y) = u\phi_x(y),$$

where $u \in \mathbb{U}_n$, $x, y \in \mathbb{B}_n$, and the map $\phi_x(\cdot)$ is defined by

$$\phi_x(y) = \omega_x\left(\frac{y - x}{1 - \langle y, x \rangle}\right), \tag{5}$$

$$\omega_x(y) = \frac{\langle y, x \rangle}{1 + \sqrt{1 - |x|^2}}x + \sqrt{1 - |x|^2}\,y.$$

The distance $d_{\mathbb{B}_n}(\cdot, \cdot)$ is given as

$$d_{\mathbb{B}_n}(x, y) = \frac{1}{2}\log\left(\frac{1 + |\phi_x(y)|}{1 - |\phi_x(y)|}\right),$$

where (by abuse of notation) $\log(\cdot)$ denotes the ordinary logarithm function.

## 3. Proposed Approach

In this section, we propose a BN algorithm on complex domains. The considered space will be denoted by $\mathcal{M}$. Mathematical proofs of our theoretical results are provided in Appendix D.

### 3.1. Computing the Batch Mean

To compute a geometric mean on $\mathcal{M}$, we use the concept of Fréchet mean. Let $x_1, \ldots, x_k$ be a set of points on $\mathcal{M}$. Then the Fréchet mean of the set is defined as

$$\mathfrak{M}_{\mathcal{M}}(x_1, \ldots, x_k) = \arg\min_{x \in \mathcal{M}} \sum_{j=1}^{k} d_{\mathcal{M}}^2(x_j, x), \tag{6}$$

where $d_{\mathcal{M}}(\cdot, \cdot)$ is a distance function on $\mathcal{M}$. For a Riemannian manifold, a common choice of $d_{\mathcal{M}}(\cdot, \cdot)$ is the Riemannian distance associated with the Riemannian metric.

When certain geometric quantities (e.g., the Riemannian exponential and logarithmic maps) are available in closed forms, the Fréchet mean can be effectively computed using Riemannian gradient descent (Absil et al., 2007). If this is not the case, then the Fréchet mean can be obtained by parameterizing $x$ on a Euclidean space and using a standard gradient descent technique to solve Eq. (6).

### 3.2. Centering and Biasing Points

BN layers in DNNs rely on two important operations[2], i.e., batch centering and biasing (Brooks et al., 2019). Those are performed via the ordinary subtraction and addition operations in Euclidean spaces. On a Riemannian manifold, meaningful analogs of subtraction and addition operations can be defined from the exponential map, logarithmic map, and parallel transport (Nguyen, 2022b; Nguyen et al., 2025b), resulting in effective Riemannian BN layers (Chen et al., 2025). However, closed forms of such geometric quantities are not always available in the general case. In this work, we propose to use automorphisms for batch centering and biasing. Note that in many complex domains, automorphisms can be characterized in the absence of the above geometric quantities (Franzoni & Vesentini, 1980).

Denote by $\phi_x(\cdot), x \in \mathcal{M}$ an automorphism of $\mathcal{M}$. Assuming that there exists an element $0_{\mathcal{M}} \in \mathcal{M}$ such that $\phi_x(x) = 0_{\mathcal{M}}$ for any $x \in \mathcal{M}$. We refer to this element as the identity element of $\mathcal{M}$. Let $m, g \in \mathcal{M}$ be the batch mean and the bias. Then for any $x \in \mathcal{M}$, subtracting $m$ from $x$ can be written as

$$\bar{x} = \phi_m(x).$$

Similarly, adding $g$ to $\bar{x}$ can be written as

$$\tilde{x} = \phi_g^{(-1)}(\bar{x}),$$

where $\phi_g^{(-1)}(\cdot)$ denotes the inverse map of $\phi_g(\cdot)$.

### 3.3. Updating the Running Mean

The new running mean is usually computed from the geodesic between the current running mean and the batch mean (Brooks et al., 2019; Chen et al., 2025). Thus, the construction for a closed form of geodesics is crucial in this step. Here we rely on *almost geodesics* instead of geodesics, and propose to use automorphisms for the construction of such curves. Assuming that we can define a scalar multiplication $t \otimes x$ on $\mathcal{M}$ where $t \in [0, 1]$ and $x \in \mathcal{M}$. Given two points $x, y \in \mathcal{M}$, we first use $\phi_x(\cdot)$ to move $x$ to $0_{\mathcal{M}}$ and $y$ to $\phi_x(y)$. We then compute an almost geodesic between $0_{\mathcal{M}}$ and $\phi_x(y)$ by solving the following equation:

$$d(0_{\mathcal{M}}, \alpha_{x,y}(t) \otimes \phi_x(y)) = t d(0_{\mathcal{M}}, \phi_x(y)), \quad (7)$$

---

[2]We do not consider the scaling operation in the present work.

---

**Algorithm 1** BN on Complex Domains

---
**Training Phase**
**Input:** Batch of $k$ points $\{x_j\}_{j=1}^k$ on $\mathcal{M}$, running mean $m_r$, bias $g$, momentum $\eta$.
$m_b \leftarrow \mathfrak{M}_{\mathcal{M}}(x_1, \ldots, x_k)$
$m_r \leftarrow \gamma_{m_r, m_b}(\eta)$
**for** $j = 1$ **to** $k$ **do**
  $\bar{x}_j \leftarrow \phi_{m_b}(x_j)$
  $\tilde{x}_j \leftarrow \phi_g^{(-1)}(\bar{x}_j)$
**end for**
**Output:** Normalized batch $\{\tilde{x}_j\}_{j=1}^k$.
**Testing Phase**
**Input:** Batch of $k$ points $\{x_j\}_{j=1}^k$ on $\mathcal{M}$, running mean $m_r$, learned bias $g$.
**for** $j = 1$ **to** $k$ **do**
  $\bar{x}_j \leftarrow \phi_{m_r}(x_j)$
  $\tilde{x}_j \leftarrow \phi_g^{(-1)}(\bar{x}_j)$
**end for**
**Output:** Normalized batch $\{\tilde{x}_j\}_{j=1}^k$.

---

where $\alpha_{x,y} : [0, 1] \to [0, 1]$, and $d(\cdot)$ is a distance function on $\mathcal{M}$. We show later that in some cases, the solution of Eq. (7) is unique and can be given in closed form.

**Definition 3.1.** Let $x, y$ be two points in $\mathcal{M}$. Then the almost geodesic $\gamma_{x,y} : [0, 1] \to \mathcal{M}$ joining $x$ and $y$ is defined as

$$\gamma_{x,y}(t) = \phi_x^{(-1)}(\gamma_{0_{\mathcal{M}}, \phi_x(y)}(t)) = \phi_x^{(-1)}(\alpha_{x,y}(t) \otimes \phi_x(y)),$$

where $\alpha_{x,y} : [0, 1] \to [0, 1]$ is a solution of Eq. (7).

We note that the concept of almost geodesic is also studied in differential geometry (Belova et al., 2024) but it is different from the one given in Definition 3.1. Our proposed method is shown in Algorithm 1.

## 4. Connection with Existing BN Layers

Our method can be seamlessly adapted to the Riemannian and Lie group settings considered in previous works (Brooks et al., 2019; Chakraborty, 2020; Lou et al., 2020; Kobler et al., 2022a;b; Chen et al., 2024; 2025). In this section, we first show the connection of the proposed BN layer with some existing BN layers. Then, we point out that our method can also be used in the framework of (2025b) to build a BN layer (RSSBN) for neural networks on Riemannian symmetric spaces (RSS). This layer has close connections with some existing ones.

### 4.1. Gyrovector Spaces

We compare our BN algorithm with those in (2019; 2025). Let $(\mathcal{M}, \oplus_g, \otimes_g)$ be a gyrovector space, where $\oplus_g$ and $\otimes_g$

are the binary operation and scalar multiplication on $\mathcal{M}$, respectively. We define the automorphism $\phi_x(\cdot), x \in \mathcal{M}$ as the left gyrotranslation (Ungar, 2014; Nguyen & Yang, 2023) by $x$, i.e.,

$$\phi_x(y) = \ominus_g x \oplus_g y,$$

where $y \in \mathcal{M}$, and $\ominus_g$ is the inverse operation on $\mathcal{M}$, i.e., $\ominus_g x \oplus_g x = 0_{\mathcal{M}}$. The map $\phi_x^{(-1)}(\cdot)$ is given by $\phi_x^{(-1)}(y) = x \oplus_g y$. Since left gyrotranslations are diffeomorphisms of $\mathcal{M}$ (Ungar, 2014; Nguyen & Yang, 2023), the automorphism is well-defined. The scalar multiplication $\otimes$ is defined as the scalar multiplication $\otimes_g$.

**Proposition 4.1.** *Let $d(\cdot, \cdot)$ be the gyrodistance (Ungar, 2014; Nguyen & Yang, 2023) defined by*

$$d(x, y) = \| \ominus_g x \oplus_g y \|_g,$$

*where $x, y \in \mathcal{M}$, and $\| \cdot \|_g$ denotes the gyrolength of the gyrovector $\ominus_g x \oplus_g y$. If $(\mathcal{M}, \oplus_g, \otimes_g)$ is an SPD gyrovector space studied in (2022b), then*

1. *The unique solution of Eq. (7) is given by $\alpha_{x,y}(t) = t$;*

2. *The almost geodesic joining any two points in $\mathcal{M}$ is the geodesic joining them.*

Therefore, for the gyrovector spaces considered in Proposition 4.1, our BN algorithm is the same as the one in (2019) and the one in (2025) without the scaling operation. In particular, our BN algorithm shares the same operations as these algorithms in three steps: centering points, biasing points, and updating the running mean.

### 4.2. Lie Groups

We now compare our BN algorithm with those in (2020; 2024) for special orthogonal groups $SO_3$. Let $L_x(\cdot)$ be the left translation by $x$. We define the automorphism $\phi_x(\cdot)$ and the scalar multiplication $\otimes$ as

$$\phi_x(y) = L_x(y), \quad t \otimes x = \exp(t \log(x)),$$

where $x, y \in SO_3$, $t \in [0, 1]$, $\exp(\cdot)$ denotes the Lie group exponential map of $SO_3$, and (by abuse of notation) $\log(\cdot)$ denotes the Lie group logarithmic map of $SO_3$. The inverse map $\phi_x^{(-1)}(\cdot)$ is then given by $\phi_x^{(-1)}(y) = L_x^{(-1)}(y)$. It is well known that $L_x(\cdot)$ is a diffeomorphism of $SO_3$. Thus the automorphism and scalar multiplication are well-defined.

**Proposition 4.2.** *Let $d(\cdot, \cdot)$ be the distance given by*

$$d(x, y) = \| \log(x^T y) \|,$$

*where $x, y \in SO_3$ and $\| \cdot \|$ is the Frobenius norm. Then the statements in Proposition 4.1 hold.*

Proposition 4.2 shows that when the scaling operations are removed from the BN algorithms in (2020; 2024), they share the same operations as our algorithm in the three steps.

### 4.3. Riemannian Symmetric Spaces

Let $\mathcal{M} = G/K$ be a Riemannian symmetric space of noncompact type, where $G$ is a connected noncompact semisimple Lie group with finite center, and $K$ is a maximal compact subgroup of $G$. Let $\mathfrak{g}$ be the Lie algebra of $G$, and let $\mathfrak{g} = \mathfrak{k} \oplus \mathfrak{p}$ be the Cartan decomposition of $\mathfrak{g}$. Denote by $\sigma : G \to \mathcal{M}, g \mapsto gK$ the natural map, $d_{0_G}\sigma : \mathfrak{g} \to T_{0_{\mathcal{M}}}\mathcal{M}$ the differential of $\sigma$ at the identity $0_G \in G$, where $T_{0_{\mathcal{M}}}\mathcal{M}$ is the tangent space of $\mathcal{M}$ at the identity $0_{\mathcal{M}} \in \mathcal{M}$. Denote by $\exp : \mathfrak{g} \to G$ the Lie group exponential map, and $\mathrm{Exp}_{0_{\mathcal{M}}} : T_{0_{\mathcal{M}}}\mathcal{M} \to \mathcal{M}$ the Riemannian exponential map. Since $\mathcal{M}$ is complete (Helgason, 1979), there exists a geodesic joining the point $0_{\mathcal{M}}$ and any point in $\mathcal{M}$. Moreover, since $d_{0_G}\sigma$ is an isomorphism and for any $u \in \mathfrak{p}$, $\mathrm{Exp}_{0_{\mathcal{M}}}(d_{0_G}\sigma(u)) = \exp(u)K$, we have $\mathcal{M} = \exp(\mathfrak{p})K$. Thus, for any $x, y \in \mathcal{M}$, there exist $u, v \in \mathfrak{p}$ such that $x = \exp(u)K, y = \exp(v)K$. We define the automorphism $\phi_x(\cdot)$ and the scalar multiplication $\otimes$ as

$$\phi_x(y) = g^{-1}hK, \quad t \otimes x = \exp(tu)K,$$

where $g = \exp(u), h = \exp(v) \in G$ and $t \in [0, 1]$. The inverse map $\phi_x^{(-1)}(\cdot)$ is then given by $\phi_x^{(-1)}(y) = ghK$.

**Proposition 4.3.** *Let $d(\cdot, \cdot)$ be the distance associated with a G-invariant Riemannian metric. Then*

1. *The unique solution of Eq. (7) is given by $\alpha_{x,y}(t) = t$;*

2. *For SPD manifolds viewed as RSS, the almost geodesic joining two points in $\mathcal{M}$ is the geodesic joining them.*

Therefore, in the case of SPD manifolds, Proposition 4.3 leads to the same conclusion drawn in Section 4.1 regarding the connection of RSSBN and the methods in (2019; 2025).

## 5. Domains in Complex Normed Spaces

We now focus on our spaces of interest which are complex domains defined by

$$\mathbb{D} = \{x \in \mathbb{E} : q(x) < 1\},$$

where $\mathbb{E}$ is a complex normed space, and $q(\cdot)$ is a continuous semi-norm on $\mathbb{E}$ (see Definition C.3).

We assume that a characterization of automorphisms on the target domain $\mathbb{D}$ has been given. Centering and biasing operations thus can be performed as explained in Section 3.2. It remains the question of how one can update the running mean. Our key idea is to perform this update by only looking at points on "straight lines" joining the current running mean $m_r$ and the batch mean $m_b$. In our case, almost geodesics will play the role of such straight lines. Thus we need to construct almost geodesics. To this aim, we first define a scalar multiplication.

On Riemannian manifolds, a method commonly adopted for defining a scalar multiplication was proposed in (2018; 2022b). However, this method is not applicable to our setting since it relies on the exponential and logarithmic maps whose closed forms (or differentiable forms) do not exist on many domains. To deal with this issue, we define the scalar multiplication using automorphisms.

**Definition 5.1.** The scalar multiplication $\otimes$ is defined as

$$t \otimes x = \phi_{0_\mathbb{D}}^{(-1)}(t\phi_{0_\mathbb{D}}(x)),$$

where $x \in \mathbb{D}$, $t \in [0, 1]$, and $0_\mathbb{D}$ is the identity element of $\mathbb{D}$.

Since $t \in [0, 1]$, we have $q(t\phi_{0_\mathbb{D}}(x)) = tq(\phi_{0_\mathbb{D}}(x))$. Thus $q(t\phi_{0_\mathbb{D}}(x)) < 1$ and the above scalar multiplication is well-defined. We can now derive an expression for almost geodesics.

**Proposition 5.2.** *Let $x, y \in \mathbb{D}$. Then the almost geodesic $\gamma_{x,y} : [0, 1] \to \mathbb{D}$ joining $x$ and $y$ is defined as*

$$\gamma_{x,y}(t) = \phi_x^{(-1)}(\alpha_{x,y}(t) \otimes \phi_x(y)),$$

*where the curve $\alpha_{x,y} : [0, 1] \to [0, 1]$ is given by*

$$q(\alpha_{x,y}(t) \otimes \phi_x(y)) = \frac{(1 + q(\phi_x(y)))^t - (1 - q(\phi_x(y)))^t}{(1 + q(\phi_x(y)))^t + (1 - q(\phi_x(y)))^t},$$

*for $t \in [0, 1]$. In particular, if for any $x \in \mathbb{D}$ and $t \in [0, 1]$, the automorphism satisfies the property that $\phi_{0_\mathbb{D}}(tx) = t\phi_{0_\mathbb{D}}(x)$, then the curve $\alpha_{x,y}$ can be given as*

$$\alpha_{x,y}(t) = \frac{1}{q(\phi_x(y))} \frac{(1 + q(\phi_x(y)))^t - (1 - q(\phi_x(y)))^t}{(1 + q(\phi_x(y)))^t + (1 - q(\phi_x(y)))^t}.$$

Given the formulae of the norm $q(\cdot)$ and the automorphism $\phi_x(\cdot)$, one can get a full description for all steps of Algorithm 1. This is the focus of the next section.

## 6. Applications

In this section, we discuss the essential components for practical implementations of BN layers on the two complex domains reviewed in Sections 2.2 and 2.3.

### 6.1. Siegel Spaces

Siegel neural networks have recently been proposed and have shown promising results in some classification tasks (Nguyen et al., 2025a). However, the authors of (2025a) only developed two building blocks for those networks, i.e., fully-connected (FC) layer and multinomial logistic regression (MLR) layers. Here we build the first BN layer for Siegel neural networks.

#### 6.1.1. GEOMETRIC MEAN ON THE SIEGEL DISK

To compute the Fréchet mean on the Siegel disk, we need to solve the optimization problem in Eq. (6). We consider the case where the distance $d_{\mathbb{SD}_n}(\cdot, \cdot)$ is used in this equation. In practice, the optimization procedure based on the formulae in Eqs. (3) and (4) suffer from numerical instability. To address this problem, we rely on other formulae of $d_{\mathbb{SD}_n}(\cdot, \cdot)$. In this section and Section 6.1.2, the map $\phi_x(\cdot), x \in \mathbb{SD}_n$ denotes the automorphism on the Siegel disk given in Eq. (2).

**Proposition 6.1.** *The distance $d_{\mathbb{SD}_n}(\cdot, \cdot)$ can be given (up to scaling) as*

$$d_{\mathbb{SD}_n}^2(x, y) = \left\| \log \left( (I + c^{\frac{1}{2}})(I - c^{\frac{1}{2}})^{-1} \right) \right\|_F^2,$$

*where $x, y \in \mathbb{SD}_n$, and $c$ is given as*

$$c = I - (I - x^H x)^{\frac{1}{2}}(I - y^H x)^{-1}(I - y^H y)(I - x^H y)^{-1} \\ (I - x^H x)^{\frac{1}{2}}, \tag{8}$$

*or, equivalently,*

$$c = \phi_x(y)^H \phi_x(y). \tag{9}$$

We note that the formula of $d_{\mathbb{SD}_n}(\cdot, \cdot)$ based on Eq. (8), which is not trivial to obtain, was first given in (2016). However, the authors did not provide the proof of this formula. We are not aware of the formula of $d_{\mathbb{SD}_n}(\cdot, \cdot)$ based on Eq. (9) in previous works.

In (2016), the authors propose a Riemannian optimization algorithm to solve Eq. (6). However, their method requires a solver for continuous Lyapunov equations which is not available in popular deep learning frameworks such as Pytorch and Tensorflow. Also, it is not clear which retraction operation they use to ensure feasibility of the manifold constraint. In our work, we parameterize the Fréchet mean on a Euclidean space and solve Eq. (6) using a standard gradient descent technique (see Appendix B.1.2).

#### 6.1.2. ALMOST GEODESICS ON THE SIEGEL DISK

Almost geodesics (see Definition 3.1) are constructed from the inverse map of $\phi_x(\cdot)$ and the solution of Eq. (7). The formula for the inverse map of $\phi_x(\cdot)$ is given below.

**Proposition 6.2.** *Let $x, y \in \mathbb{SD}_n$. Then the inverse map $\phi_x^{(-1)}(\cdot)$ is given by*

$$\phi_x^{(-1)}(y) = (I - xx^H)^{\frac{1}{2}}(I + yx^H)^{-1}(y + x)(I - x^H x)^{-\frac{1}{2}}.$$

Note that $\mathbb{SD}_n$ can be defined as

$$\mathbb{SD}_n = \{x \in \text{Sym}_{n,\mathbb{C}} : q(x) < 1\},$$

*Table 1.* Results (mean accuracy ± standard deviation) computed over 5 runs for radar clutter classification. The tuple $(C, N)$ below each dataset indicates the number of classes $C$ and the size of the dataset $N$. The dimension of the time series is set to 30.

| Method | Dataset 1 (D1) (20, 950) | Dataset 2 (D2) (40, 450) | Dataset 3 (D3) (60, 650) | Dataset 4 (D4) (80, 900) | Dataset 5 (D5) (100, 1000) | Dataset 6 (D6) (120, 1200) |
|---|---|---|---|---|---|---|
| ComplexLSTM-CLN | 26.62±0.39 | 26.53±0.85 | 27.82±0.39 | 47.82±3.53 | 19.75±0.90 | 24.92±0.46 |
| kNN (Cabanes & Nielsen, 2021) | 28.16±0.0 | 31.81±0.0 | 36.15±0.0 | 51.33±0.0 | 30.68±0.0 | 29.61±0.0 |
| Kernel-Siegel (Chevallier et al., 2016) | 27.83±0.0 | 33.05±0.0 | 35.01±0.0 | 52.15±0.0 | 32.80±0.0 | 28.31±0.0 |
| SiegelNetFC (Nguyen et al., 2025a) | 31.24±0.44 | 41.16±0.55 | 40.02±0.48 | 62.47±0.51 | 42.86±0.22 | 34.22±0.46 |
| SiegelNetBN (Ours) | **33.83±0.34** | **43.29±0.40** | **41.38±0.37** | **65.93±0.32** | **45.61±0.12** | **37.10±0.26** |

where $q(x) = \|x\|_2$ and $\|\cdot\|_2$ denotes the spectral norm of a matrix given by its largest singular value. Using Proposition 5.2, we deduce the following result.

**Corollary 6.3.** *Let $x, y \in \mathbb{SD}_n$. Then the almost geodesic $\gamma_{x,y} : [0, 1] \to \mathbb{SD}_n$ joining $x$ and $y$ is given by*

$$\gamma_{x,y}(t) = \phi_x^{(-1)}\left(\frac{1}{\|z\|_2}\frac{(1 + \|z\|_2)^t - (1 - \|z\|_2)^t}{(1 + \|z\|_2)^t + (1 - \|z\|_2)^t}z\right),$$

*where $z = \phi_x(y)$.*

### 6.2. The Complex Unit Ball

It has been shown (Franzoni & Vesentini, 1980) that the map $\phi_x(\cdot)$ defined in Eq. (5) satisfies

$$\phi_{-x}(\phi_x(y)) = y,$$

for any $x, y \in \mathbb{B}_n$. Thus the inverse map $\phi_x^{(-1)}(\cdot)$ is precisely the map $\phi_{-x}(\cdot)$. The formula of almost geodesics can be given in Corollary 6.4.

**Corollary 6.4.** *Let $x, y \in \mathbb{B}_n$. Then the almost geodesic $\gamma_{x,y} : [0, 1] \to \mathbb{B}_n$ joining $x$ and $y$ is given by*

$$\gamma_{x,y}(t) = \phi_x^{(-1)}\left(\frac{1}{|z|}\frac{(1 + |z|)^t - (1 - |z|)^t}{(1 + |z|)^t + (1 - |z|)^t}z\right),$$

*where $z = \phi_x(y)$, and the map $\phi_x(\cdot)$ is given in Eq. (5).*

## 7. Related Works

Several works have proposed extensions of BN layers in the Riemannian setting. The work in (2019) was among the first which introduced BN layers for SPD neural networks. Since these layers only use the sample mean to normalize data, the sample variance was then added to the normalization operation to provide better control on the sample statistics (Kobler et al., 2022a;b; Chakraborty, 2020). Extensions of BN layers were also developed for general Riemannian manifolds (Lou et al., 2020), Lie groups (Chen et al., 2024), and gyrogroups (Chen et al., 2025). All these works assume either the availability of closed forms for some geometric

quantities (e.g., the exponential map) or a special algebraic structure of the considered space which are not required in our work. Furthermore, none of these works leverages the concept of Kobayashi pseudodistance to build almost geodesics for the update of the running mean as done in the present work.

## 8. Experiments

This section presents our experimental results on the tasks of radar clutter classification and node classification. More experimental details and evaluations are given in Appendix B.

### 8.1. Radar Clutter Classification

We consider the task of radar clutter classification which aims at recognizing different types of radar clutter using information recorded by a radar related to seas, forests, fields, cities and other environmental elements surrounding the radar (Cabanes, 2022). Following (2022; 2025a), we assume that radar signals are stationary centered autoregressive (AR) Gaussian time series (Barbaresco, 1996; Billingsley, 2002; Barbaresco, 2013). The AR model is given by

$$u_t + \sum_{j=1}^r c_j u_{t-j} = v_t,$$

where $r$ $(r > 1)$ is the order of the AR model, $u_t \in \mathbb{C}_n$ is the vector of signals at time $t$, $c_j \in \mathbb{C}^{n \times n}, j = 1, \ldots, r$ are the prediction coefficients (AR parameters), and $v_t \in \mathbb{C}_n$ is the prediction error at time $t$ which is assumed to be a multidimensional Gaussian random variable. The method in (2022; 2025a) is used to generate the input data for our network from time series. Each input of our network can be represented as $(p^0, x^1, \ldots, x^{r-1})$, where $p^0 \in \mathrm{Sym}_n^+$ and $x^1, \ldots, x^{r-1} \in \mathbb{SD}_n$, which belongs to $\mathrm{Sym}_n^+ \times \mathbb{SD}_n^{r-1}$.

We create an architecture consisting of three layers. Our proposed BN layer is used as the first layer. Given a batch of $k$ points $\{(p_j^0, x_j^1, \ldots, x_j^{r-1})\}_{j=1}^k$, where $(p_j^0, x_j^1, \ldots, x_j^{r-1}) \in \mathrm{Sym}_n^+ \times \mathbb{SD}_n^{r-1}$, the BN layer is applied to each batch of $k$ points $\{x_j^l\}_{j=1}^k, l = 1, \ldots, r-1$, resulting in a batch of $k$ points $\{(p_j^0, \tilde{x}_j^1, \ldots, \tilde{x}_j^{r-1})\}_{j=1}^k$.

*Table 2.* Computation times (seconds) per batch w.r.t. the dimension of time series (Hardware: Intel(R) Xeon(R) Gold 6230R CPU @ 2.10GHz).

| | Dimension | 30 | 60 | 90 | 120 | 150 |
|---|---|---|---|---|---|---|
| Train | SiegelNetFC (Nguyen et al., 2025a) | 0.0442 | 0.1070 | 0.1717 | 0.2113 | 0.4383 |
| | SiegelNetBN (Ours) | 0.1906 | 0.3370 | 0.4246 | 0.4669 | 0.6779 |
| | Scale factor | × 4.31 | × 3.14 | × 2.47 | × 2.20 | × 1.54 |
| Test | SiegelNetFC (Nguyen et al., 2025a) | 0.0022 | 0.0047 | 0.0122 | 0.0192 | 0.0455 |
| | SiegelNetBN (Ours) | 0.0134 | 0.0253 | 0.0517 | 0.0677 | 0.0964 |
| | Scale factor | × 6.09 | × 5.38 | × 4.23 | × 3.52 | × 2.11 |

*Table 3.* Results (mean accuracy $\pm$ standard deviation) computed over 10 runs for node classification. The hyperbolicity values of Airport, Pubmed, and Cora datasets are 1, 3.5, and 11, respectively. A lower hyperbolicity value means more hyperbolic.

| Dimension | Dataset | Airport | Pubmed | Cora |
|---|---|---|---|---|
| 8 | HNN-RBN-H (Chen et al., 2025) | 62.34±0.68 | 62.50±2.36 | 29.63±7.55 |
| | HNN-GyroBN-H (Chen et al., 2025) | 70.38±2.40 | 61.46±3.63 | 41.33±2.94 |
| | CBallNetBN (Ours) | **74.42±3.43** | **71.22±0.76** | **58.46±1.11** |
| 16 | HNN-RBN-H (Chen et al., 2025) | 63.15±0.76 | 66.18±1.84 | 40.00±3.46 |
| | HNN-GyroBN-H (Chen et al., 2025) | 72.38±0.87 | 65.67±1.14 | 42.06±0.78 |
| | CBallNetBN (Ours) | **79.58±0.86** | **72.73±0.56** | **60.88±1.49** |
| 32 | HNN-RBN-H (Chen et al., 2025) | 60.49±2.90 | 67.81±2.31 | 41.29±4.80 |
| | HNN-GyroBN-H (Chen et al., 2025) | 78.79±2.01 | 67.19±2.33 | 43.43±1.99 |
| | CBallNetBN (Ours) | **80.24±1.35** | **73.16±0.46** | **61.08±1.17** |

The second layer (ICAYLEY) applies the inverse matrix Cayley transformation to the components of the input belonging to $\mathbb{SD}_n$. The last layer is the $\text{QMLR}_{\text{Sym}_n^+ \times \mathbb{SH}_n^{r-1}}$ layer proposed in (2025a) for classification. We use the Kähler distance $d_{\mathbb{SD}_n}(\cdot, \cdot)$ for the computation of the Fréchet mean. Our network is compared against the following baselines: (1) ComplexLSTM-CLN which is a variant of a vanilla LSTM with layer normalization (Ba et al., 2016) for complex-valued data; (2) k-Nearest Neighbors (kNN) based on the Kähler distance (Cabanes & Nielsen, 2021); (3) Kernel-Siegel based on kernel density estimation on Siegel spaces (Chevallier et al., 2016); and (4) SiegelNetFC which consists of the ICAYLEY layer, the AFC layer (an FC layer) proposed in (2025a), and the $\text{QMLR}_{\text{Sym}_n^+ \times \mathbb{SH}_n^{r-1}}$ layer. Note that this is the best architecture in (2025a). We rename it here for convenience of presentation. SiegelNetFC uses the same input as our network.

Results in Tab. 1 show that our network surpasses its competitors in terms of mean accuracy on all the datasets. In particular, SiegelNetBN improves SiegelNetFC by 2.59%, 2.13%, 1.35%, 3.46%, 2.75%, and 2.88% on datasets 1, 2, 3, 4, 5, and 6, respectively. This shows the effectiveness of our proposed BN layer compared to FC layers in (2025a). This also indicates that in the considered setting, almost geodesics constructed from the Kobayashi pseudodistance in the Siegel disk are good approximations of geodesics in this domain. Therefore, our results suggest that almost geodesics can be used in other applications which require a closed form for geodesics on Siegel spaces. It is noted that both kNN and Kernel-Siegel based on traditional learning models are significantly outperformed by our network. This demonstrates that neural networks on Siegel spaces are promising tools for analysis of complex signals. The computation times of SiegelNetBN and SiegelNetFC w.r.t. different dimensions of the time series are given in Tab. 2. While SiegelNetBN is more time-consuming than SiegelNetFC, mainly due to the computation of the Fréchet mean in the BN layer, the former still scales well with the dimensionality of input data.

## 8.2. Node Classification

We validate the proposed BN layer on the complex unit ball by performing node classification experiments on three datasets, i.e., Airport (Zhang & Chen, 2018), Pubmed (Namata et al., 2012), and Cora (Sen et al., 2008), each of them contains a single graph with thousands of labeled nodes. We use HNN-GyroBN-H (Chen et al., 2025) as the baseline. Since many building blocks for hyperbolic neural networks (HNNs) have been developed (Shimizu et al., 2021; Bdeir et al., 2024), the focus of this experiment is to compare our proposed BN layer against BN layers for HNNs, and we do not aim to build a state-of-the-art HNN architecture for the considered task. The encoder of HNN-GyroBN-H has a number of blocks, each of which is composed of a linear layer (HypLinear), a BN layer (GBN), and an activation layer (HypAct) built on the Poincaré ball. Our network CBallNetBN is constructed by replacing the GBN layer with our BN layer. Specifically, CBallNetBN converts the output of the HypLinear layer to $\mathbb{B}_n$ using the Discrete Fourier Transform (DFT) (Xiao et al., 2022), then applies our BN layer, and finally converts its output to the Poincaré ball using the inverse DFT. We also test HNN-RBN-H (Chen et al., 2025) which uses the BN layer (RBN) proposed in (2020) instead of GBN. Results in Tab. 3 show that our BN layer outperforms both the GBN and RBN layers by large margins. These results suggest that neural networks built on the complex unit ball can potentially improve those built on the (real) Poincaré ball. The computation times of HNN-GyroBN-H and CBallNetBN are reported in Tab. 4. It is noted that the scale factor of the computation times of these networks gets smaller when the dimension of node embeddings increases.

## 8.3. Ablation Study

**Impact of the BN layer in SiegelNetBN** We remove the BN layer from SiegelNetBN and evaluate the performance of the resulting network SiegelNetNoBN on the task of radar clutter classification. Results are given in Tab. 5 (see

Table 4. Computation times (seconds) per epoch w.r.t. the dimension of node embeddings on Airport dataset (Hardware: Intel(R) Core(TM) i7-8565U CPU @ 1.80GHz).

| Dimension | 8 | 16 | 32 |
|---|---|---|---|
| HNN-GyroBN-H (Chen et al., 2025) | 0.0291 | 0.0335 | 0.0536 |
| CBallNetBN (Ours) | 0.0725 | 0.0740 | 0.0951 |
| Scale factor | $\times$ 2.49 | $\times$ 2.21 | $\times$ 1.77 |

Table 5. Impact of the BN layer in SiegelNetBN. Results (mean accuracy $\pm$ standard deviation) are computed over 5 runs.

| Method | D1 | D2 | D3 | D4 |
|---|---|---|---|---|
| SiegelNetNoBN | 29.37±0.48 | 38.93±0.51 | 36.56±0.45 | 62.53±0.54 |
| SiegelNetBN | **33.83±0.34** | **43.29±0.40** | **41.38±0.37** | **65.93±0.32** |

Appendix B.1.3 for results on all the datasets). As can be observed, our BN layer exhibits benefit similar to BN layers in DNNs, i.e., it leads to better accuracies and more stable results across all the datasets. In terms of mean accuracy, SiegelNetBN improves SiegelNetNoBN by 4.45%, 4.35%, 4.82%, and 3.40% on D1, D2, D3, and D4, respectively.

**Impact of the BN layer in CBallNetBN** We conduct the same ablation study for CBallNetBN by removing the BN layer from it and evaluate the performance of the resulting network CBallNetNoBN on the task of node classification. The embedding dimension is set to 16. Results in Tab. 6 again confirm the efficacy of the proposed method. In terms of mean accuracy, CBallNetBN improves CBallNetNoBN by 7.21%, 1.15%, and 2.59% on Airport, Pubmed, and Cora datasets, respectively.

## 9. Complexity Analysis

**The proposed BN algorithm** We analyze the computational costs in the training stages of our BN layer and the well-established Riemannian BN layer (SPDBN) in (Brooks et al., 2019). Denote by $n_{iters}$ the number of iterations for Fréchet mean computation, $k$ the batch size, $n$ the dimension of input matrices. The steps of our BN layer have the following time complexities:

- Fréchet mean computation: $O(n_{iters}kn^3)$

- Running mean update: $O(n^3)$

- Centering and biasing points: $O(kn^3)$

Overall, our BN layer has a time complexity of order $O(n_{iters}kn^3)$ which is the time complexity of the SPDBN layer. Here we assume that $n_{iters}$ is the number of iterations for Karcher mean computation in the SPDBN layer.

Table 6. Impact of the BN layer in CBallNetBN. Results (mean accuracy $\pm$ standard deviation) are computed over 10 runs. The embedding dimension is set to 16.

| Dataset | Airport | Pubmed | Cora |
|---|---|---|---|
| Hyperbolicity $\delta$ | $\delta = 1$ | $\delta = 3.5$ | $\delta = 11$ |
| CBallNetNoBN | 72.36±1.18 | 71.58±1.06 | 58.29±0.83 |
| CBallNetBN | **79.58±0.86** | **72.73±0.56** | **60.88±1.49** |

**Distance computation** The computations for both the Kähler and Kobayashi distances have the same time complexity using unoptimized implementations. More specifically, they have the following time complexities:

- The computation of the Kähler distance based on Eq. (8) or Eq. (9) has a time complexity of order $O(n^3)$.

- The computation of the Kobayashi distance has a time complexity of order $O(n^3)$.

## 10. Limitation of the Proposed Approach

One of the key assumptions of our approach is the availability of a closed form of automorphisms on the considered domain. Therefore, it is not applicable to settings in which such a closed form does not exist.

A major computational bottleneck in our BN algorithm is the computation of the Fréchet mean. Therefore, an interesting direction for future work is to develop efficient algorithms for computing the Fréchet mean on complex domains.

Similar to the networks in (2021; 2025a), our BN layer for Siegel neural networks is built upon operations on Siegel spaces which are generally expensive. Like SPD neural networks (Huang & Gool, 2017; Brooks et al., 2019), Siegel neural networks are heavily based on the SVD operation which suffers from high computational cost when dealing with high-dimensional input. Also, low-dimensional embeddings on complex domains are not able to capture complex relationships within data which can affect the performance of our method.

## 11. Conclusion

In this paper, we introduce a novel BN layer for neural networks on complex domains. We discuss the connection of the proposed BN layer with some existing BN layers on Riemannian manifolds and Lie groups. We derive the essential components for practical implementations of BN layers on the Siegel disk domain and the complex unit ball. Finally, we provide experimental evaluations showing that our approach has the potential to improve existing approaches on the radar clutter classification and node classification tasks.

## Acknowledgements

We are grateful for the constructive comments and feedback from the anonymous reviewers.

## Impact Statement

This paper presents work whose goal is to advance the field of Machine Learning. There are many potential societal consequences of our work, none which we feel must be specifically highlighted here.

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

*Table 7.* The main notations used in our paper.

| Symbol | Name |
|---|---|
| $\mathbb{C}$ | Set of complex numbers |
| $\mathbb{C}_n$ | $n$-dimensional complex vector space |
| $\mathcal{M}$ | General space |
| $\mathbb{E}$ | Complex normed space |
| $\mathbb{D}$ | Complex domain |
| $0_{\mathbb{D}}$ | Identity element of $\mathbb{D}$ |
| $I_n$ | $n \times n$ identity matrix |
| $\mathbf{0}_n$ | $n \times n$ zero matrix |
| $q(\cdot)$ | Continuous semi-norm on $\mathbb{E}$ |
| $\mathbb{B}$ | Open unit disc in $\mathbb{C}$ |
| $\mathbb{B}_n$ | Complex unit ball in $\mathbb{C}_n$ $(n > 1)$ |
| $g_{\mathbb{D}}^K(\cdot, \cdot)$ | Kobayashi differential metric on $\mathbb{D}$ |
| $d_{\mathbb{D}}^K(\cdot, \cdot)$ | Kobayashi pseudodistance on $\mathbb{D}$ |
| $\tilde{g}_{\mathbb{D}}(\cdot, \cdot)$ | Integrated form of $g_{\mathbb{D}}(\cdot, \cdot)$ |
| $\mathrm{Hol}(\mathbb{D}_1, \mathbb{D}_2)$ | Set of holomorphic maps from $\mathbb{D}_1$ to $\mathbb{D}_2$ |
| $\mathbb{SH}_n$ | Siegel upper half space of $n \times n$ matrices |
| $\mathbb{SD}_n$ | Siegel disk of $n \times n$ matrices |
| $\mathrm{GL}_n$ | Group of $n \times n$ invertible matrices |
| $\mathrm{Sp}_n$ | Real symplectic group of $n \times n$ matrices |
| $\mathrm{O}_n$ | Group of $n \times n$ orthogonal matrices |
| $\mathrm{SO}_n$ | Group of $n \times n$ special orthogonal matrices |
| $\mathrm{U}_n$ | Space of $n \times n$ unitary matrices |
| $\mathrm{Sym}_n$ | Space of $n \times n$ real symmetric matrices |
| $\mathrm{Sym}_{n,\mathbb{C}}$ | Space of $n \times n$ complex symmetric matrices |
| $\mathrm{Sym}_n^+$ | Space of $n \times n$ SPD matrices |
| $\mathrm{H}_n^+$ | Space of $n \times n$ HPD matrices |
| $\phi_x(\cdot)$ | Automorphism |
| $\varphi(\cdot)$ | Matrix Cayley transformation |
| $\gamma_{x,y}(\cdot)$ | Almost geodesic joining $x$ and $y$ |

## A. Notations

The main notations used in our paper are summarized in Tab. 7.

## B. Experiments

### B.1. Radar Clutter Classification

#### B.1.1. DATASETS AND EXPERIMENTAL SETTINGS

Our datasets are simulated using the method in (2022). Given the length $N$ of a time series, we generate the first $r$ temporal elements using the following equation:

$$y = b^{\frac{1}{2}}x,$$

where $b$ is a Block-Toeplitz HPD matrix and $x$ is a standard complex Gaussian random vector whose dimension is equal to the product of the dimension of the time series and its length. The $N - r$ remaining elements are then generated using the

*Table 8.* Statistics of our synthetic datasets.

| Parameter | Dimension | Length | Number of classes | Order | Size |
|-----------|-----------|--------|-------------------|-------|------|
| Dataset 1 | 30 | 50 | 20 | 3 | 950 |
| Dataset 2 | 30 | 50 | 40 | 2 | 450 |
| Dataset 3 | 30 | 50 | 60 | 3 | 650 |
| Dataset 4 | 30 | 50 | 80 | 4 | 900 |
| Dataset 5 | 30 | 50 | 100 | 2 | 1000 |
| Dataset 6 | 30 | 50 | 120 | 2 | 1200 |

following equation:

$$u_t + \sum_{j=1}^{r} c_j u_{t-j} = v_t,$$

where $u_t \in \mathbb{C}_n$ is the vector of signals at time $t$, $c_j \in \mathbb{C}^{n \times n}, j = 1, \ldots, r$ are the prediction coefficients (AR parameters), and $v_t \in \mathbb{C}_n$ is the prediction error at time $t$ which is assumed to be a multidimensional Gaussian random variable. We use the same matrix $b$ to simulate time series of the same class. The simulation of our datasets is based on the settings given in Tab. 8.

### B.1.2. OPTIMIZATION AND HYPERPARAMETERS

Algorithm 2 shows the computation of the product space representation $(\tilde{p}^0, x^1, \ldots, x^{r-1}) \in \mathbb{H}_n^+ \times \mathbb{SD}_n^{r-1}$ for a time series. We convert a point in $\mathbb{H}_n^+ \times \mathbb{SD}_n^{r-1}$ to $\mathrm{Sym}_n^+ \times \mathbb{SD}_n^{r-1}$ by taking the real part of the component of the point belonging to $\mathbb{H}_n^+$ and then convert it to $\mathrm{Sym}_n^+$. To convert a real $n \times n$ matrix $u$ to $\mathrm{Sym}_n$, we set $u = \frac{1}{2}(u + u^T)$. To convert a matrix $u \in \mathrm{Sym}_n$ to $\mathrm{Sym}_n^+$, we compute the eigendecomposition of $u$ as $u = kdk^{-1}$ where $k \in \mathrm{O}_n$ (the group of $n \times n$ orthogonal matrices) and $d$ is a $n \times n$ diagonal matrix, and set $u = kd_\epsilon k^{-1}$ where

$$(d_\epsilon)_{ii} = \begin{cases} d_{ii} & \text{if } d_{ii} > \epsilon \\ \epsilon & \text{otherwise} \end{cases}$$

The value of $\epsilon$ is set to $1e-4$ in our experiments. We use the above transformations to enforce the manifold constraint for data in $\mathrm{Sym}_n^+ \times \mathbb{SD}_n^{r-1}$ and $\mathrm{Sym}_n^+ \times \mathbb{SH}_n^{r-1}$. The Kähler distance $d_{\mathbb{SD}_n}(\cdot, \cdot)$ is computed using Eq. (8).

We parameterize the Fréchet mean $x$ of a set of points $x_1, \ldots, x_k \in \mathbb{SD}_n$ on a Euclidean space. Let $y = \varphi^{(-1)}(x) = u + iv \in \mathbb{SH}_n$ where $u \in \mathrm{Sym}_n$ and $v \in \mathrm{Sym}_n^+$ (see Section 2.2.3 for the definition of the map $\varphi(\cdot)$). Then both $u$ and $v$ can be parameterized using vectors in $\mathbb{R}^{\frac{n(n+1)}{2}}$. Hence, we have that $\varphi^{(-1)}(x) = \varphi_1(a) + i\varphi_2(b)$, where the maps $\varphi_1(\cdot)$ and $\varphi_2(\cdot)$ are given as

$$\varphi_1(a) = \frac{1}{2}\left(\mathrm{mat}(a) + \mathrm{mat}(a)^T\right), \quad \varphi_2(a) = \exp\left(\frac{1}{2}\left(\mathrm{mat}(a) + \mathrm{mat}(a)^T\right)\right),$$

where $a \in \mathbb{R}^{\frac{n(n+1)}{2}}$, $\mathrm{mat}(a)$ transforms $a$ into a $n \times n$ lower triangular matrix, and (by abuse of notation) $\exp(\cdot)$ denotes the matrix exponential. Eq. (6) now becomes

$$(a, b) = \operatorname*{arg\,min}_{a,b \in \mathbb{R}^{n(n+1)/2}} \sum_{j=1}^{k} d_{\mathbb{SD}_n}^2(x_j, \varphi(\varphi_1(a) + i\varphi_2(b))). \tag{10}$$

We use the methods in (2021; 2025a) to compute the eigenvalues and inverses of points on Siegel spaces. For the $\mathrm{QMLR}_{\mathrm{Sym}_n^+ \times \mathbb{SH}_n^{r-1}}$ layer, the probability of class $l$ is computed as

$$p(y = l|x) \propto \exp\left(\frac{\left|\sum_{j=0}^{r-1} \langle \log(h_{j,l}^{-1} g_j g_j^T h_{j,l}^{-T}), \log(w_{j,l} w_{j,l}^T) \rangle\right|}{\sqrt{\sum_{j=0}^{r-1} \left\|\log(w_{j,l} w_{j,l}^T)\right\|^2}}\right),$$

---

**Algorithm 2** Compute the representation of a time series in $\mathbb{H}_n^+ \times \mathbb{SD}_n^{r-1}$

---

**Input:** A vector sequence $u_0, \ldots, u_{N-1} \in \mathbb{C}_n$
$f_{0,k} \leftarrow b_{0,k} \leftarrow u_k, k = 0, \ldots, N-1$
$\tilde{p}_0 \leftarrow \frac{1}{N} \sum_{k=0}^{N-1} u_k u_k^H$
**for** $i = 1$ **to** $r - 1$ **do**
$\quad r_{i-1}^f \leftarrow \sum_{k=i}^{N-1} f_{i-1,k} f_{i-1,k}^H$
$\quad r_{i-1}^b \leftarrow \sum_{k=i}^{N-1} b_{i-1,k-1} b_{i-1,k-1}^H$
$\quad r_{i-1}^{fb} \leftarrow \sum_{k=i}^{N-1} f_{i-1,k} b_{i-1,k-1}^H$
$\quad w_i \leftarrow -(r_{i-1}^f)^{-\frac{1}{2}} r_{i-1}^{fb} \left( (r_{i-1}^b)^{-\frac{1}{2}} \right)^H$
$\quad \begin{cases} f_{i,k} \leftarrow f_{i-1,k} + w_i b_{i-1,k-1} & k = i, \ldots, N-1 \\ b_{i,k} \leftarrow b_{i-1,k-1} + w_i^H f_{i-1,k} & k = i, \ldots, N-1 \end{cases}$
**end for**
**Output:** $(\tilde{p}^0, x^1, \ldots, x^{r-1}) \in \mathbb{H}_n^+ \times \mathbb{SD}_n^{r-1}$

---

where $x = (p^0, \bar{x}^1, \ldots, \bar{x}^{r-1}) \in \mathrm{Sym}_n^+ \times \mathbb{SH}_n^{r-1}, p^0 = g_0 \, \mathrm{O}_n \in \mathrm{Sym}_n^+, g_0 \in \mathrm{GL}_n, \bar{x}^j = g_j \, \mathrm{SpO}_{2n} \in \mathbb{SH}_n, g_j \in \mathrm{Sp}_{2n}, j = 1, \ldots, r-1$, and $h_{0,l}, w_{0,l} \in \mathrm{GL}_n, h_{j,l}, w_{j,l} \in \mathrm{Sp}_{2n}, j = 1, \ldots, r-1$ are the parameters associated with class $l, l = 1, \ldots, M$. Here $\mathrm{GL}_n$ denotes the general linear group. Please refer to Appendix C.3 for the definitions of $\mathrm{Sp}_{2n}$ and $\mathrm{SpO}_{2n}$. Note that $g_j, j = 1, \ldots, r-1$ are computed from $\bar{x}^j$ as in Eq. (12), and $h_{j,l}, w_{j,l}, j = 1, \ldots, r-1$ have the same form as $g_j$.

The implementation of ComplexLSTM-CLN is based on the ComplexNN toolbox. We tested different loss functions (Yadav & Jerripothula, 2023; Barrachina et al., 2023) for complex-valued input but obtained poor results. Therefore, we concatenate the real and imaginary parts of the output of ComplexLSTM-CLN and feed the resulting representation to a linear layer (with real-valued input) for classification. The number of layers and the hidden size are set to 2 and 20, respectively. kNN uses the distance given (Jeuris & Vandebril, 2016) by

$$d_{kNN}^2(x,y) = r \left\| \log \left( (x^0)^{-\frac{1}{2}} y^0 (x^0)^{-\frac{1}{2}} \right) \right\|^2 + \sum_{j=1}^{r-1} \frac{r-j}{4} \mathrm{Tr} \left( \log^2 \left( \left( I + c_j^{\frac{1}{2}} \right) \left( I - c_j^{\frac{1}{2}} \right)^{-1} \right) \right),$$

where $x = (x^0, x^1, \ldots, x^{r-1}), y = (y^0, y^1, \ldots, y^{r-1}) \in \mathbb{H}_n^+ \times \mathbb{SD}_n^{r-1}$, and $c_j, j = 1, \ldots, r-1$ are given by

$$c_j = (y^j - x^j) \left( I - (x^j)^H y^j \right)^{-1} \left( (y^j)^H - (x^j)^H \right) \left( I - x^j (y^j)^H \right)^{-1}.$$

We perform 10-fold cross-validation with the training set to find the best value of $k$. For Kernel-Siegel which is based on kernel density estimation (KDE), we use the method in (2017) and the one in (2016) for KDEs on $\mathrm{Sym}_n^+$ and $\mathbb{SD}_n$, and then combine them to obtain a KDE on the product space $\mathrm{Sym}_n^+ \times \mathbb{SD}_n^{r-1}$. We use the quartic kernel (Chevallier et al., 2016) given by

$$K(x) = \frac{3}{\pi} (1 - x^2)^2 \mathbf{1}_{x<1}.$$

Our networks are implemented in the Pytorch framework and trained using cross-entropy loss and Adadelta optimizer for 2000 epochs. The learning rate and the batch size are set to $1e-2$ and 25, respectively. For ComplexLSTM-CLN, we use Adam optimizer and set the learning rate to $1e-3$. The number of iterations for computing the Fréchet mean is set to 5. Results are averaged over 5 random parameter initializations for each model. All experiments are performed using a machine with an Intel(R) Xeon(R) W-2223 CPU @ 3.60GHz. To measure computation times, we use a machine with an Intel(R) Xeon(R) Gold 6230R CPU @ 2.10GHz.

### B.1.3. MORE RESULTS

Tab. 9 presents the results from Tab. 1 of the main paper along with those of SiegelNetKobayashiBN, which is trained by replacing the Kähler distance $d_{\mathbb{SD}_n}(\cdot, \cdot)$ with the Kobayashi distance $d_{\mathbb{SD}_n}^K(\cdot, \cdot)$ for the computation of the Fréchet mean

*Table 9.* Results (mean accuracy $\pm$ standard deviation) computed over 5 runs for radar clutter classification. The tuple below each dataset indicates the number of classes and the size of the dataset.

| Method | Dataset 1 (D1) (20, 950) | Dataset 2 (D2) (40, 450) | Dataset 3 (D3) (60, 650) | Dataset 4 (D4) (80, 900) | Dataset 5 (D5) (100, 1000) | Dataset 6 (D6) (120, 1200) |
|---|---|---|---|---|---|---|
| ComplexLSTM-CLN | 26.62±0.39 | 26.53±0.85 | 27.82±0.39 | 47.82±3.53 | 19.75±0.90 | 24.92±0.46 |
| kNN (Cabanes & Nielsen, 2021) | 28.16±0.0 | 31.81±0.0 | 36.15±0.0 | 51.33±0.0 | 30.68±0.0 | 29.61±0.0 |
| Kernel-Siegel (Chevallier et al., 2016) | 27.83±0.0 | 33.05±0.0 | 35.01±0.0 | 52.15±0.0 | 32.80±0.0 | 28.31±0.0 |
| SiegelNetFC (Nguyen et al., 2025a) | 31.24±0.44 | 41.16±0.55 | 40.02±0.48 | 62.47±0.51 | 42.86±0.22 | 34.22±0.46 |
| SiegelNetBN | 33.83±0.34 | **43.29±0.40** | 41.38±0.37 | **65.93±0.32** | **45.61±0.12** | 37.10±0.26 |
| SiegelNetKobayashiBN | **34.26±0.31** | 43.02±0.44 | **42.15±0.33** | 65.24±0.35 | 44.92±0.14 | **37.92±0.24** |

*Table 10.* Impact of the BN layer in SiegelNetBN. Results (mean accuracy $\pm$ standard deviation) are computed over 5 runs.

| Method | D1 | D2 | D3 | D4 | D5 | D6 |
|---|---|---|---|---|---|---|
| SiegelNetNoBN | 29.37±0.48 | 38.93±0.51 | 36.56±0.45 | 62.53±0.54 | 40.18±0.19 | 32.68±0.42 |
| SiegelNetBN | **33.83±0.34** | **43.29±0.40** | **41.38±0.37** | **65.93±0.32** | **45.61±0.12** | **37.10±0.26** |

in the BN layer of SiegelNetBN. As can be observed, SiegelNetKobayashiBN gives results competitive to SiegelNetBN. This further demonstrates that almost geodesics constructed from the Kobayashi pseudodistance in the Siegel disk are good approximations of geodesics in this domain. Tab. 10 shows the effectiveness of the BN layer of SiegelNetBN on all the datasets. It can be noted that the BN layer improves the performance of SiegelNetBN in all cases.

## B.2. Node Classification

### B.2.1. DATASETS AND EXPERIMENTAL SETTINGS

**Airport (Zhang & Chen, 2018)** It is a flight network dataset from OpenFlights.org where nodes represent airports, edges represent the airline Routes, and node labels are the populations of the country where the airport belongs.

**Pubmed (Namata et al., 2012)** It is a standard benchmark describing citation networks where nodes represent scientific papers in the area of medicine, edges are citations between them, and node labels are academic (sub)areas.

**Cora (Sen et al., 2008)** It is a citation network where nodes represent scientific papers in the area of machine learning, edges are citations between them, and node labels are academic (sub)areas. Each publication in the dataset is described by a 0/1-valued word vector indicating the absence/presence of the corresponding word from the dictionary.

Our experiments are based on the experimental settings in (Chami et al., 2019). Specifically, we use 70/15/15 percent splits for Airport dataset and standard splits with 20 train examples per class for Pubmed and Cora datasets.

### B.2.2. OPTIMIZATION AND HYPERPARAMETERS

We parameterize the Fréchet mean of a set of points in $\mathbb{B}_n$ on a Euclidean space (the real and imaginary parts). The networks are implemented in the Pytorch framework and trained using cross-entropy loss and Adam optimizer. CBallNetBN projects the result obtained after each transformation between the complex and real domains to the target domain. The results of HNN-GyroBN-H and HNN-RBN-H are obtained using their official implementation[3]. Hyperparameter settings are based on the work of (2019). The number of epochs and the learning rate are set to $5000$ and $1e-2$, respectively. We use early stopping based on validation set performance with a patience of 100 epochs. The weight decay for Airport dataset is set to 0. The weight decays for Pubmed and Cora datasets are set to $1e-4$ and $1e-3$, respectively. The embedding dimension is set to 16. The number of blocks (see Section 8.2) in the encoder of the networks is set to 2. Results are averaged over 10 random parameter initializations on the final test set for each model. All experiments are performed using a machine with an Intel(R) Core(TM) i7-8565U CPU @ 1.80GHz.

---

[3] https://github.com/GitZH-Chen/GyroBN.

## B.3. Action Recognition

To further demonstrate the applicability of our method, we conduct action recognition experiments on NTU60 (Shahroudy et al., 2016) and NTU120 (Liu et al., 2019) datasets. We focus on comparing neural networks on different Riemannian manifolds. This allows us to investigate the efficacy of embedding action data into different Riemannian manifolds for the task of action recognition. Here we consider a challenging setting in which the input data contain only one of the three coordinates (channels) of the 3D human joint positions. We choose some state-of-the-art SPD neural networks as baselines since they have proven to be effective in comparison with other Riemannian neural networks on the considered task (Huang et al., 2018; Nguyen & Yang, 2023). In particular, we compare SiegelNetBN and SiegelNetKobayashiBN against the following baselines: (1) SPDNet (Huang & Gool, 2017)[4]; (2) SPDNetBN (Brooks et al., 2019)[5]; (3) LieBN (Chen et al., 2024)[6]; (4) GBWBN (Wang et al., 2025)[7]; and SiegelNetFC. GBWBN is a variant of SPDNetBN in which the BN layer is built upon the Generalized Bures-Wasserstein metric (Han et al., 2023).

### B.3.1. Datasets and Experimental Settings

**NTU60**  It has 56880 sequences of 3D skeleton data with 60 classes. Each frame contains the 3D coordinates of 25 or 50 body joints. We use the mutual (interaction) actions for classification (11 classes). For the cross-subject (XSubject60) protocol (Shahroudy et al., 2016), this results in 7319 and 3028 sequences for training and testing, respectively.

**NTU120**  It has 114480 sequences in 120 action classes, captured by 106 subjects with three cameras views. Each frame contains the 3D coordinates of 25 or 50 body joints. We use the mutual (interaction) actions for classification (26 classes). For the cross-subject (XSubject120) protocol (Liu et al., 2019), the 106 subjects are split into training and testing groups where each group contains 53 subjects. This results in 13072 and 11660 sequences for training and testing, respectively. For the cross-setup (XSetup120) protocol (Liu et al., 2019), training data contain samples with even setup IDs, and testing data contain samples with odd setup IDs. This results in 11864 and 12868 sequences for training and testing, respectively.

### B.3.2. Optimization and Hyperparameters

For the baselines, the input data are computed as in (Huang & Gool, 2017; Brooks et al., 2019). The Bimap layers output SPD matrices having the same size as input matrices. For LieBN, we test the models based on the Affine-Invariant metric (AIM) (Pennec, 2006) and the Log-Cholesky metric (LCM) (Lin, 2019) which have shown to give the best performances on the considered task (Chen et al., 2024). The baselines have the same architectures proposed in the original papers (Huang & Gool, 2017; Brooks et al., 2019; Chen et al., 2024; Wang et al., 2025). We use the method in Section B.1.2 to compute the input data of our networks. Each input of our networks belongs to the product space $\mathrm{Sym}_n^+ \times \mathbb{SD}_n$.

We use the same method in Section B.1.2 for optimizing parameters. All networks are trained using cross-entropy loss and Adadelta optimizer for 2000 epochs. The learning rate and the batch size are set to $1e-2$ and 256, respectively. Results are computed over 5 random parameter initializations for each model. All experiments are performed using a machine with an Intel(R) Xeon(R) W-2223 CPU @ 3.60GHz.

### B.3.3. Results

Results in Tab. 11 show that our networks achieve the best mean accuracies and standard deviations on all the datasets. Also, the performance gaps of our networks and SPD models are significant in most cases. These results reveal that in some scenarios, embedding Euclidean data into Siegel spaces could be beneficial for classification tasks. Similar to the results in Tab. 9, SiegelNetKobayashiBN achieves the same level of performance as SiegelNetBN. This again confirms the effectiveness of the Kobayashi pseudodistance for building BN layers in Siegel neural networks.

### B.3.4. Improving Neural Networks on Grassmann Manifolds

In this section, we show how to improve the performance of neural networks on Grassmann manifolds using our proposed BN layer. We use GrNet (Huang et al., 2018) which is a state-of-the-art neural network on Grassmann manifolds as the

---

[4] https://github.com/zhiwu-huang/SPDNet.
[5] https://papers.nips.cc/paper/2019/hash/6e69ebbfad976d4637bb4b39de261bf7-Abstract.html.
[6] https://github.com/GitZH-Chen/LieBN.git.
[7] https://github.com/jjscc/GBWBN.

*Table 11.* Comparison of SPD and Siegel neural networks. Results (mean accuracy $\pm$ standard deviation) computed over 5 runs for action recognition on NTU60 and NTU120 datasets. XSubject60, XSubject120, and XSetup120 correspond to the cross-subject setting of NTU60, the cross-subject setting of NTU120, and the cross-setup setting of NTU120, respectively.

| Method | XSubject60 | | | XSubject120 | | | XSetup120 | | |
|---|---|---|---|---|---|---|---|---|---|
| | x-channel | y-channel | z-channel | x-channel | y-channel | z-channel | x-channel | y-channel | z-channel |
| SPDNet (Huang & Gool, 2017) | 63.96±0.32 | 61.69±0.29 | 58.25±0.54 | 47.90±0.34 | 43.57±0.30 | 42.04±0.44 | 48.42±0.39 | 42.80±0.46 | 40.46±0.42 |
| SPDNetBN (Brooks et al., 2019) | 64.97±0.26 | 62.66±0.28 | 59.77±0.52 | 49.70±0.30 | 46.60±0.33 | 42.37±0.49 | 50.13±0.34 | 45.33±0.41 | 45.13±0.45 |
| LieBN-AIM-(1.5) (Chen et al., 2024) | 63.91±0.30 | 61.04±0.28 | 58.55±0.56 | 47.80±0.31 | 43.81±0.31 | 42.29±0.40 | 48.97±0.36 | 43.11±0.44 | 42.22±0.47 |
| LieBN-LCM-(0.5) (Chen et al., 2024) | 65.18±0.41 | 63.16±0.35 | 59.65±0.58 | 49.12±0.37 | 45.41±0.36 | 40.50±0.53 | 49.57±0.43 | 45.51±0.55 | 42.39±0.49 |
| GBWBN (Wang et al., 2025) | 32.83±0.0 | 37.58±0.0 | 33.44±0.0 | 16.47±0.0 | 16.60±0.0 | 9.32±0.0 | 18.37±0.0 | 19.36±0.0 | 12.69±0.0 |
| SiegelNetFC (Nguyen et al., 2025a) | 67.21±0.34 | 65.18±0.31 | 61.43±0.55 | 52.08±0.35 | 48.24±0.36 | 44.02±0.51 | 53.04±0.40 | 48.37±0.48 | 45.32±0.49 |
| SiegelNetBN | **69.25±0.21** | **67.25±0.24** | 63.60±0.38 | **55.20±0.22** | 50.64±0.29 | 45.10±0.31 | **56.28±0.28** | 50.45±0.35 | 46.48±0.37 |
| SiegelNetKobayashiBN | 68.81±0.23 | 67.03±0.20 | **64.34±0.35** | 55.47±0.20 | **50.93±0.30** | **46.27±0.34** | 55.76±0.31 | **51.28±0.29** | **46.80±0.33** |

*Table 12.* Improving neural networks on Grassmann manifolds using the proposed BN layer. Results (mean accuracy $\pm$ standard deviation) computed over 5 runs for action recognition on NTU60 and NTU120 datasets. XSubject60, XSubject120, and XSetup120 correspond to the cross-subject setting of NTU60, the cross-subject setting of NTU120, and the cross-setup setting of NTU120, respectively.

| Method | XSubject60 | | | XSubject120 | | | XSetup120 | | |
|---|---|---|---|---|---|---|---|---|---|
| | x-channel | y-channel | z-channel | x-channel | y-channel | z-channel | x-channel | y-channel | z-channel |
| GrNet (Huang et al., 2018) | 54.89±0.24 | 49.68±0.26 | 49.69±0.46 | 35.73±0.21 | 32.16±0.36 | 29.81±0.17 | 38.42±0.31 | 33.40±0.22 | 32.51±0.25 |
| GrNetSiegelBN | **67.03±0.20** | **62.38±0.22** | **58.85±0.39** | **48.49±0.18** | **42.66±0.29** | **39.78±0.14** | **49.83±0.26** | **44.56±0.19** | **42.05±0.17** |

baseline. Let $x = (p^0, x^1) \in \mathrm{Sym}_n^+ \times \mathbb{SH}_n$ be an input of the $\mathrm{QMLR}_{\mathrm{Sym}_n^+ \times \mathbb{SH}_n}$ layer of SiegelNetBN. Let $g_{x^1}$ be the point in $\mathrm{Sp}_{2n}$ which transforms $iI$ to $x^1$ (see Appendix C.3.1), that is

$$g_{x^1} = \begin{bmatrix} v^{\frac{1}{2}} & uv^{-\frac{1}{2}} \\ \mathbf{0}_n & v^{-\frac{1}{2}} \end{bmatrix},$$

where $x^1 = u + iv$. Let $h \in \mathrm{Sp}_{2n}$ be a parameter of the $\mathrm{QMLR}_{\mathrm{Sym}_n^+ \times \mathbb{SH}_n}$ layer given by

$$h = \begin{bmatrix} v_1^{\frac{1}{2}} & u_1 v_1^{-\frac{1}{2}} \\ \mathbf{0}_n & v_1^{-\frac{1}{2}} \end{bmatrix},$$

where $u_1 \in \mathrm{Sym}_n$ and $v_1 \in \mathrm{Sym}_n^+$. GrNet uses a Euclidean FC layer as the final layer for classification. We vectorize the lower part of the matrix $\log(h^{-1} g_{x^1} g_{x^1}^T h^{-T})$, then concatenate it with the input feature vector of the FC layer of GrNet, and finally feed the resulting representation to a Euclidean FC layer for classification. The network based on this pipeline is called GrNetSiegelBN. Note that GrNetSiegelBN does not rely on $p^0$ which is part of the input data of SiegelNetBN belonging to $\mathrm{Sym}_n^+$. Thus, compared to GrNet, GrNetSiegelBN only uses additional features on Siegel spaces learned by our BN layer. Results of the two networks are given in Tab. 12. It can be observed that our proposed BN layer significantly improves the performance of GrNet. In particular, GrNetSiegelBN improves GrNet by more than 9% in terms of mean accuracy in all cases.

## C. Definitions and Basic Facts

In this section, we present definitions and basic facts used in our paper. For greater mathematical detail and in-depth discussion, we refer the interested reader to (1980).

### C.1. Invariant Distances

We review the Poincaré distance (Helgason, 1979) on the unit disc which is the starting point for studying invariant distances on complex domains.

**Definition C.1 (Poincaré differential metric (Franzoni & Vesentini, 1980)).** Let $\mathbb{B}$ be the open unit disc in $\mathbb{C}$, i.e.,

$$\mathbb{B} = \{x \in \mathbb{C} : |x| < 1\}.$$

Let $x \in \mathbb{B}$ and $v \in \mathbb{C}$. Then the Poincaré differential metric $g_{\mathbb{B}} : \mathbb{B} \times \mathbb{C} \to [0, +\infty)$ is defined as

$$g_{\mathbb{B}}(x, v) = \frac{|v|}{1 - |x|^2}.$$

The Poincaré distance can then be given as follows.

**Definition C.2** (**Poincaré distance (Franzoni & Vesentini, 1980)**). The Poincaré distance $d_{\mathbb{B}}(\cdot, \cdot)$ is the integrated form of the Poincaré differential metric on $\mathbb{B}$ and is given as

$$d_{\mathbb{B}}(x, y) = \frac{1}{2} \log \left( \frac{1 + \left| \frac{x-y}{1-x\bar{y}} \right|}{1 - \left| \frac{x-y}{1-x\bar{y}} \right|} \right),$$

where $x, y \in \mathbb{B}$.

## C.2. Complex Spaces

**Definition C.3** (**Normed spaces**). Let $\mathbb{E}$ be a vector space over a field $\mathbb{K}$, where $\mathbb{K} = \mathbb{R}$ or $\mathbb{C}$. A function $q : \mathbb{E} \to \mathbb{R}$ is a norm on $\mathbb{E}$ if

(1) $q(x) \geq 0$ for all $x \in \mathbb{E}$;

(2) $q(\alpha x) = |\alpha| q(x)$ for $x \in \mathbb{E}$ and $\alpha \in \mathbb{K}$;

(3) $q(x - y) \leq q(x - z) + q(z - y)$ for all $x, y, z \in \mathbb{E}$; and

(4) $q(x) = 0$ if and only if $x = 0$.

If the final axiom does not necessarily hold we call $q$ a semi-norm. The pair $(\mathbb{E}, q(\cdot))$ is called a normed space. A normed space is said to be strictly normed if

(5) $q(x + y) = q(x) + q(y)$ implies that $y = tx$ for some $t > 0$, or else $x = 0$.

The Kobayashi pseudodistance can also be defined via the concept of analytic chain.

**Definition C.4** (**Analytic chain (Franzoni & Vesentini, 1980)**). Let $\mathbb{D}$ be a domain in a complex normed space $\mathbb{E}$, and let $x, y \in \mathbb{D}$. An analytic chain joining $x$ and $y$ in $\mathbb{D}$ consists of $2m$ points $z_1', z_1'', \ldots, z_m', z_m''$ in $\mathbb{B}$ and of $m$ functions $f_j \in \mathrm{Hol}(\mathbb{B}, \mathbb{D})$ such that

$$f_1(z_1') = x, \ldots, f_j(z_j'') = f_{j+1}(z_{j+1}'),$$

for $j = 1, \ldots, m - 1, f_m(z_m'') = y$.

**Definition C.5** (**The Kobayashi pseudodistance (Franzoni & Vesentini, 1980)**). Let $\mathbb{D}$ be a domain in a complex normed space $\mathbb{E}$, and let $x, y \in \mathbb{D}$. Then the Kobayashi pseudodistance $\bar{d}_{\mathbb{D}}^K(x, y)$ is defined as

$$\bar{d}_{\mathbb{D}}^K(x, y) = \inf\{d_{\mathbb{B}}(z_1', z_1'') + d_{\mathbb{B}}(z_2', z_2'') + \ldots + d_{\mathbb{B}}(z_m', z_m'')\},$$

where the infimum is taken over all choices of analytic chains joining $x$ and $y$ in $\mathbb{D}$.

It has been proved (Franzoni & Vesentini, 1980) that the above definition of the Kobayashi pseudodistance is equivalent to Definition 2.5 in the main paper, i.e.,

$$\bar{d}_{\mathbb{D}}^K(x, y) = d_{\mathbb{D}}^K(x, y),$$

where $x, y \in \mathbb{D}$.

## C.3. Siegel Spaces

In this section, we further discuss the geometries of the Siegel upper half space and the Siegel disk that have not provided in the main paper.

## C.3.1. THE SIEGEL UPPER HALF SPACE

The Siegel upper half space has the structure of a homogeneous space (Siegel, 1943). Let $\mathrm{Sp}_{2n}$ be the real symplectic group defined by

$$\mathrm{Sp}_{2n} = \left\{ s \in \mathbb{R}^{2n \times 2n} | s^T e_{2n} s = e_{2n} \right\}, \text{ with } e_{2n} = \begin{bmatrix} \mathbf{0}_n & I \\ -I & \mathbf{0}_n \end{bmatrix},$$

where $\mathbf{0}_n$ denotes the $n \times n$ zero matrix. The left action of $\mathrm{Sp}_{2n}$ on $\mathbb{SH}_n$ is defined by the generalized linear fractional transformation (Siegel, 1943) given by

$$\begin{aligned} \psi : \mathrm{Sp}_{2n} \times \mathbb{SH}_n &\to \mathbb{SH}_n \\ g[x] &\mapsto (ax + b)(cx + d)^{-1}, \end{aligned} \tag{11}$$

where $g = \begin{bmatrix} a & b \\ c & d \end{bmatrix} \in \mathrm{Sp}_{2n}$. This action is transitive. The isotropy subgroup of the element $iI \in \mathbb{SH}_n$ is given by

$$\mathrm{SpO}_{2n} = \left\{ \begin{bmatrix} a & b \\ -b & a \end{bmatrix} : a^T a + b^T b = I, a^T b = b^T a \right\} = \mathrm{Sp}_{2n} \cap \mathrm{O}_{2n}.$$

This gives rise to the following identification

$$\mathbb{SH}_n \cong \mathrm{Sp}_{2n} / \mathrm{SpO}_{2n}.$$

The element in $\mathrm{Sp}_{2n}$ that transforms $iI$ to $x = u + iv \in \mathbb{SH}_n$ via the group action is given by

$$g_{u+iv} = \begin{bmatrix} v^{\frac{1}{2}} & uv^{-\frac{1}{2}} \\ \mathbf{0}_n & v^{-\frac{1}{2}} \end{bmatrix}. \tag{12}$$

The Riemannian metric of the Siegel upper half space is a generalization of the case $n = 1$. In the case $n = 1$, the Siegel upper half space becomes the upper half-plane and the linear fractional transformation in Eq. (11) is given by

$$y = \frac{ax + b}{cx + d},$$

where $a, b, c, d \in \mathbb{R}$ and $ad - bc = 1$. In this case, there exists a transformation mapping two given points $x, y$ of the upper half-plane into two other given points $x_1, y_1$ of the upper half-plane, if and only if

$$R(x, y) = R(x_1, y_1),$$

where $R(x, y)$ denotes the cross-ratio given by

$$R(x, y) = \frac{x - y}{x - \bar{y}} \frac{\bar{x} - \bar{y}}{\bar{x} - y}.$$

To generalize the above result to the Siegel upper half space, one considers the cross-ratio $R(\cdot, \cdot)$ given as

$$R(x, y) = (x - y)(x - \bar{y})^{-1}(\bar{x} - \bar{y})(\bar{x} - y)^{-1},$$

where $x, y \in \mathbb{SH}_n$. Let $g = \begin{bmatrix} a & b \\ c & d \end{bmatrix} \in \mathrm{Sp}_{2n}$, $x_1 = g[x]$, and $y_1 = g[y]$. Then

$$R(x, y) = (cy + d)^T R(x_1, y_1) \left( (cy + d)^T \right)^{-1},$$

which shows that $R(x, y)$ and $R(x_1, y_1)$ have the same eigenvalues. Now, taking the second differential of $R(x, y)$ at the point $y = x$, one gets

$$d^2 R(x, y) = \frac{1}{2} dx v^{-1} d\bar{x} v^{-1},$$

where $x = u + iv$. Therefore

$$d^2 \operatorname{Tr}(R(x,y)) = \operatorname{Tr}(d^2 R(x,y)) = \frac{1}{2} \operatorname{Tr}(v^{-1} dx v^{-1} d\bar{x}), \tag{13}$$

where the first equality is due to the community of the differential and the trace operator.

The Hermitian differential form in Eq. (13) is the Riemannian metric of the Siegel upper half space. It is invariant under the group action of $\operatorname{Sp}_{2n}$ on $\mathbb{SH}_n$ since $\operatorname{Tr}(R(x,y)) = \operatorname{Tr}(R(g[x], g[y]))$ for any $g \in \operatorname{Sp}_{2n}$. The resulting Riemannian distance $d_{\mathbb{SH}_n}(x,y)$ is given (Siegel, 1943) by

$$d_{\mathbb{SH}_n}(x,y) = \sqrt{\sum_{j=1}^{n} \log^2 \left( \frac{1 + r_j^{\frac{1}{2}}}{1 - r_j^{\frac{1}{2}}} \right)},$$

where $r_j, j = 1, \ldots, n$ are the eigenvalues of the cross-ratio $R(x,y)$.

### C.3.2. THE SIEGEL DISK

The Kähler metric in the Siegel disk is given (Barbaresco, 2013; Jeuris & Vandebril, 2016) by

$$ds_x^2 = \operatorname{Tr} \left( (I - xx^H)^{-1} dx (I - x^H x)^{-1} dx^H \right),$$

where $x \in \mathbb{SD}_n$.

The (left) action of the real symplectic group $\operatorname{Sp}_{2n}$ on $\mathbb{SD}_n$ is obtained via the inverse matrix Cayley transformation which converts points in $\mathbb{SD}_n$ to $\mathbb{SH}_n$. This matrix is given by

$$c = \begin{bmatrix} iI & iI \\ -I & I \end{bmatrix}.$$

Let $g \in \operatorname{Sp}_{2n}$ and let $c^{-1} g c = \begin{bmatrix} a_0 & b_0 \\ c_0 & d_0 \end{bmatrix}$. Then the matrix $g_0 = \begin{bmatrix} a_0 & b_0 \\ \bar{b}_0 & \bar{a}_0 \end{bmatrix} \in \operatorname{Sp}_{2n}$ acts isometrically on $\mathbb{SD}_n$ by the following action:

$$g_0[x] = (a_0 x + b_0)(\bar{b}_0 x + \bar{a}_0)^{-1},$$

where $x \in \mathbb{SD}_n$.

The Kobayashi distance $d^K_{\mathbb{SD}_n}(\cdot, \cdot)$ is given (Jeuris & Vandebril, 2016) by

$$d^K_{\mathbb{SD}_n}(x,y) = \frac{1}{2} \log \left( \frac{1 + \|\phi_x(y)\|_2}{1 - \|\phi_x(y)\|_2} \right), \tag{14}$$

where $x, y \in \mathbb{SD}_n$.

## D. Mathematical Proofs

### D.1. Proof of Proposition 4.1

*Proof.* Let $x, y, z \in \mathcal{M}$. Then

$$\begin{aligned}
d(\phi_x(y), \phi_x(z)) &= d(\ominus_g x \oplus_g y, \ominus_g x \oplus_g z) \\
&= \| \ominus_g (\ominus_g x \oplus_g y) \oplus_g (\ominus_g x \oplus_g z) \|_g \\
&= \| \operatorname{gyr}[\ominus_g x, y](\ominus_g y \oplus_g z) \|_g \\
&= \| \ominus_g y \oplus_g z \|_g \\
&= d(y, z),
\end{aligned}$$

where the third equation follows from the Left Gyrotranslation Theorem (Ungar, 2014), and the fourth equation follows from the invariance of the norm under gyrations (Ungar, 2014).

In the case of SPD gyrovector spaces studied in (2022b), one has

$$
\begin{aligned}
d(0_{\mathcal{M}}, \alpha_{x,y}(t) \otimes \phi_x(y)) &= \| \log(\alpha_{x,y}(t) \otimes \phi_x(y)) \|_g \\
&= \alpha_{x,y}(t) \| \log(\phi_x(y)) \|_g \\
&= \alpha_{x,y}(t) d(0_{\mathcal{M}}, \phi_x(y)),
\end{aligned}
$$

where (by abuse of notation) $\log(\cdot)$ is the Riemannian logarithmic map at the identity, the first and the third equations follow from the definition of the SPD gyrodistance and that of the SPD inner product (see (2023), Definition 2.15), and the second equation follows from the definition of the scalar multiplication on $\mathcal{M}$.

It is immediate that the unique solution of Eq. (7) is given by $\alpha_{x,y}(t) = t$. According to Definition 3.1, the almost geodesic joining $x$ and $y$ is given by

$$
\phi_x^{(-1)}(\alpha_{x,y}(t) \otimes \phi_x(y)) = x \oplus_g t \otimes_g (\ominus_g x \oplus_g y).
$$

It can be easily seen that geodesics on SPD gyrovector spaces in (2022b) can be expressed as the right-hand side of the above equation. This completes the proof of the proposition.

$\square$

### D.2. Proof of Proposition 4.2

*Proof.* One has

$$
\begin{aligned}
d(0_{\mathrm{SO}_n}, \alpha_{x,y}(t) \otimes \phi_x(y)) &= \| \log(\alpha_{x,y}(t) \otimes \phi_x(y)) \| \\
&= \| \alpha_{x,y}(t) \log(\phi_x(y)) \| \\
&= \alpha_{x,y}(t) \| \log(\phi_x(y)) \| \\
&= \alpha_{x,y}(t) d(0_{\mathrm{SO}_n}, \phi_x(y)),
\end{aligned}
$$

where the first and the last equations follow from the definition of the distance on $\mathrm{SO}_3$, the second equation follows from the definition of the scalar multiplication on $\mathrm{SO}_3$, and the third equation follows from the fact that $\alpha_{x,y}(t) \in [0, +\infty)$ for $t \in [0, 1]$.

It is immediate that the unique solution of Eq. (7) is given by $\alpha_{x,y}(t) = t$. According to Definition 3.1, the almost geodesic joining $x$ and $y$ is given by

$$
\begin{aligned}
\phi_x^{(-1)}(\alpha_{x,y}(t) \otimes \phi_x(y)) &= L_x^{(-1)}(t \otimes L_x(y)) \\
&= x \exp(t \log(x^{-1} y)).
\end{aligned}
$$

This completes the proof of the proposition.

$\square$

### D.3. Proof of Proposition 4.3

*Proof.* To prove the first statement, we need a result from (2009).

**Lemma D.1.** *Let $\overline{\mathfrak{a}^+}$ be the closed Weyl chamber. Let the map (Cartan projection) $\mu : G \to \overline{\mathfrak{a}^+}$ be determined by $g = k \exp(\mu(g))k'$ with $g \in G$ and $k, k' \in K$. The map $\mu(\cdot)$ is a continuous, proper, surjective map (Helgason, 1979). Denote by $o$ the origin $K$ in $X$. Let $\rho : X \to \overline{\mathfrak{a}^+}$ be the map sending $x = g[o] \in X$ to $\mu(g)$, where $g \in G$. Then for all $x, x' \in X$,*

$$
\| \rho(x) - \rho(x') \| \leq d(x, x').
$$

*Moreover, if $x, x' \in \exp(\overline{\mathfrak{a}^+})[o]$, then $d(x, x') = \| \rho(x) - \rho(x') \|$.*

By the definition of the scalar multiplication,

$$
\alpha(t) \otimes \exp(u)K = \exp(\alpha(t)u)K.
$$

Let $\phi_x(y) = \exp(u)K$ where $u \in \mathfrak{p}$. Then

$$
\begin{aligned}
d(0_{\mathcal{M}}, \phi_x(y)) &= d(o, \exp(u)K) \\
&= \|\rho(o) - \rho(\exp(u)[o])\| \\
&= \|\mu(\exp(u))\|,
\end{aligned}
$$

where the second equation follows from Lemma D.1. Let $\mathrm{Ad}_G$ be the adjoint representation of $G$. Since $u \in \mathfrak{p}$, there exists $k \in K$ and $a \in \overline{\mathfrak{a}^+}$ such that $u = \mathrm{Ad}(k)a$. Thus $\mu(\exp(u)) = a$ and therefore

$$
d(0_{\mathcal{M}}, \phi_x(y)) = \|a\|.
$$

Also, one has

$$
\begin{aligned}
d(0_{\mathcal{M}}, \alpha_{x,y}(t) \otimes \phi_x(y)) &= d(0_{\mathcal{M}}, \exp(\alpha_{x,y}(t)u)K) \\
&= d(o, \exp(\alpha_{x,y}(t)u)K) \\
&= \|\rho(o) - \rho(\exp(\alpha_{x,y}(t)u)[o])\| \\
&= \|\mu(\exp(\alpha_{x,y}(t)u))\| \\
&= \|\alpha_{x,y}(t)a\|,
\end{aligned}
$$

where the third equation follows from Lemma D.1 and the last equation follows from the fact that $u = \mathrm{Ad}(k)a$. Eq. (7) now becomes

$$
\|\alpha_{x,y}(t)a\| = t\|a\|,
$$

which has the unique solution given by $\alpha_{x,y}(t) = t$.

To prove the second statement, we need the following lemma.

**Lemma D.2.** *Let $a, b \in \mathrm{Sym}_n^+$. Then there exists $k \in \mathrm{O}_n$ such that $abk \in \mathrm{Sym}_n^+$.*

*Proof.* We shall show that $k = b^{-1}a^{-1}(ab^2a)^{\frac{1}{2}}$ satisfies the condition of the lemma. First, note that

$$
kk^T = b^{-1}a^{-1}(ab^2a)^{\frac{1}{2}}(ab^2a)^{\frac{1}{2}}a^{-1}b^{-1} = I.
$$

Also, one has

$$
k^T k = (ab^2a)^{\frac{1}{2}}a^{-1}b^{-1}b^{-1}a^{-1}(ab^2a)^{\frac{1}{2}} = I.
$$

Furthermore, $abk = (ab^2a)^{\frac{1}{2}}$ and it is easy to see that $ab^2a$ is symmetric positive definite since $ab^2a = (ab^2a)^T$ and for any $x \in \mathbb{R}^n \setminus \{\mathbf{0}\}$, $xab^2ax^T = (xab)(xab)^T > 0$. $\qquad \square$

When $\mathcal{M}$ is $\mathrm{Sym}_n^+$, $G$ is the general linear group $\mathrm{GL}_n$ (with entries in $\mathbb{R}$) and $K$ is $\mathrm{O}_n$. It has been shown (Helgason, 1979) that the Cartan decomposition of $\mathfrak{g}$ is given by

$$
\mathfrak{g} = \mathfrak{o}_n \oplus \mathrm{Sym}_n,
$$

where $\mathfrak{o}_n$ denotes the Lie algebra of $\mathrm{O}_n$. Since $\mathfrak{p} \cong \mathrm{Sym}_n$, the automorphism $\phi_x(\cdot)$ is given by

$$
\phi_x(y) = x^{-\frac{1}{2}} y^{\frac{1}{2}} K.
$$

By Lemma D.2, there exists $k \in K$ such that $z = x^{-\frac{1}{2}} y^{\frac{1}{2}} k \in \mathrm{Sym}_n^+$. Let $z = \exp(v), v \in \mathfrak{p}$. Then

$$
\begin{aligned}
\alpha_{x,y}(t) \otimes \phi_x(y) &= t \otimes x^{-\frac{1}{2}} y^{\frac{1}{2}} K \\
&= t \otimes \exp(v)K \\
&= \exp(tv)K.
\end{aligned}
$$

Therefore, one has

$$\phi_x^{(-1)}(\alpha_{x,y}(t) \otimes \phi_x(y)) = x^{\frac{1}{2}} \exp(tv)K.$$

The almost geodesic $\gamma_{x,y}$ joining $x$ and $y$ is thus given by

$$\phi_x^{(-1)}(\alpha_{x,y}(t) \otimes \phi_x(y)) = x^{\frac{1}{2}} \exp(2tv)x^{\frac{1}{2}}$$
$$= x^{\frac{1}{2}}(x^{-\frac{1}{2}}yx^{-\frac{1}{2}})^t x^{\frac{1}{2}},$$

where the second equation follows from the fact that $x^{-\frac{1}{2}}y^{\frac{1}{2}}K = \exp(v)K$ shown above. This concludes the proof of the proposition.

$\square$

### D.4. Proof of Proposition 5.2

*Proof.* We need the following result (Franzoni & Vesentini, 1980).

**Theorem D.3.** *Let $q(\cdot)$ be a continuous semi-norm on $\mathbb{E}$, and let*

$$\mathbb{D} = \{x \in \mathbb{E} : q(x) < 1\}.$$

*Then*

$$d_{\mathbb{D}}^K(\mathbf{0}, x) = d_{\mathbb{B}}(0, q(x)),$$

*where $x \in \mathbb{D}$, and $\mathbf{0}$ is the image of the holomorphic map $\tau : \mathbb{B} \to \mathbb{D}, v \mapsto \frac{v}{q(x)}x$ for $q(x) > 0$ and $v = 0$.*

For domains considered in our paper, we can assume that $0_{\mathbb{D}} = \mathbf{0}$. We rely on the Kobayashi pseudodistance in Eq. (7) to construct almost geodesics. Using the result in Theorem D.3, Eq. (7) now becomes

$$d_{\mathbb{B}}(0, q(\alpha_{x,y}(t) \otimes \phi_x(y))) = t d_{\mathbb{B}}(0, q(\phi_x(y))).$$

We thus have

$$\frac{1}{2} \log \frac{1 + |q(\alpha_{x,y}(t) \otimes \phi_x(y))|}{1 - |q(\alpha_{x,y}(t) \otimes \phi_x(y))|} = t \frac{1}{2} \log \frac{1 + |q(\phi_x(y))|}{1 - |q(\phi_x(y))|}.$$

Since $q(x) \in [0, +\infty)$ for all $x \in \mathbb{E}$,

$$\frac{1}{2} \log \frac{1 + q(\alpha_{x,y}(t) \otimes \phi_x(y))}{1 - q(\alpha_{x,y}(t) \otimes \phi_x(y))} = t \frac{1}{2} \log \frac{1 + q(\phi_x(y))}{1 - q(\phi_x(y))},$$

which results in

$$\frac{1 + q(\alpha_{x,y}(t) \otimes \phi_x(y))}{1 - q(\alpha_{x,y}(t) \otimes \phi_x(y))} = \left(\frac{1 + q(\phi_x(y))}{1 - q(\phi_x(y))}\right)^t.$$

Solving the above equation leads to

$$q(\alpha_{x,y}(t) \otimes \phi_x(y)) = \frac{(1 + q(\phi_x(y)))^t - (1 - q(\phi_x(y)))^t}{(1 + q(\phi_x(y)))^t + (1 - q(\phi_x(y)))^t}.$$

By the definition of the scalar multiplication $\otimes$ and the assumption on the automorphism,

$$\alpha_{x,y}(t) \otimes \phi_x(y) = \phi_{0_{\mathbb{D}}}^{(-1)}(\alpha_{x,y}(t)\phi_{0_{\mathbb{D}}}(\phi_x(y)))$$
$$= \phi_{0_{\mathbb{D}}}^{(-1)}(\phi_{0_{\mathbb{D}}}(\alpha_{x,y}(t)\phi_x(y)))$$
$$= \alpha_{x,y}(t)\phi_x(y).$$

Therefore

$$q(\alpha_{x,y}(t)\phi_x(y)) = \frac{(1 + q(\phi_x(y)))^t - (1 - q(\phi_x(y)))^t}{(1 + q(\phi_x(y)))^t + (1 - q(\phi_x(y)))^t}.$$

By axiom (2) in Definition C.3 and the fact that $\alpha_{x,y}(t) \geq 0$ for $t \in [0, 1]$, we have

$$\alpha_{x,y}(t)q(\phi_x(y)) = \frac{(1 + q(\phi_x(y)))^t - (1 - q(\phi_x(y)))^t}{(1 + q(\phi_x(y)))^t + (1 - q(\phi_x(y)))^t},$$

which implies that

$$\alpha_{x,y}(t) = \frac{1}{q(\phi_x(y))} \frac{(1 + q(\phi_x(y)))^t - (1 - q(\phi_x(y)))^t}{(1 + q(\phi_x(y)))^t + (1 - q(\phi_x(y)))^t}.$$

One can verify that $\alpha_{x,y}(t) \in [0, 1]$ for $t \in [0, 1]$. This completes the proof of the proposition.

$\square$

### D.5. Proof of Proposition 6.1

We first prove the following results.

**Lemma D.4.** *Let* $x \in \mathbb{SD}_n$. *Then*

$$x = (I - xx^H)^{-\frac{1}{2}} x (I - x^H x)^{\frac{1}{2}}.$$
$$(I - xx^H)^{-1} x = x(I - x^H x)^{-1}$$
$$(I - xx^H)^{-1} - x(I - x^H x)^{-1} x^H = I$$

*Proof.* Let $x = udv$, where $d$ is a diagonal matrix with real diagonal entries, and $u$ and $v$ are unitary matrices. Then

$$(I - xx^H)^{-\frac{1}{2}} x (I - x^H x)^{\frac{1}{2}} = u(I - d^2)^{-\frac{1}{2}} u^{-1} udvv^{-1}(I - d^2)^{\frac{1}{2}} v$$
$$= udv$$
$$= x,$$

which proves the first identity. Also, note that

$$x(I - x^H x) = x - xx^H x \tag{15}$$
$$= (I - xx^H)x,$$

or, equivalently,

$$(I - xx^H)^{-1} x = x(I - x^H x)^{-1},$$

which proves the second identity. By Eq. (15),

$$x = (I - xx^H) x (I - x^H x)^{-1},$$

Therefore

$$I - xx^H = I - (I - xx^H)x(I - x^H x)^{-1} x^H,$$

which results in

$$I = (I - xx^H)^{-1} - x(I - x^H x)^{-1} x^H.$$

$\square$

We also need the following lemma.

**Lemma D.5.** *Let* $x, y \in \mathbb{SD}_n$. *Then*

$$I - \phi_x(y)^H \phi_x(y) = (I - x^H x)^{\frac{1}{2}} (I - y^H x)^{-1} (I - y^H y)(I - x^H y)^{-1} (I - x^H x)^{\frac{1}{2}}.$$

*Proof.* Let z = $\phi_x(y)$. Then

$$I - z^H z = (I - x^H x)^{\frac{1}{2}}(I - x^H x)^{-1}(I - x^H x)^{\frac{1}{2}} - (I - x^H x)^{\frac{1}{2}}(I - y^H x)^{-1}(y^H - x^H)(I - xx^H)^{-1}(y - x)$$
$$(I - x^H y)^{-1}(I - x^H x)^{\frac{1}{2}}.$$

Therefore

$$I - z^H z = (I - x^H x)^{\frac{1}{2}}(I - y^H x)^{-1}A(I - x^H y)^{-1}(I - x^H x)^{\frac{1}{2}}, \tag{16}$$

where $A = (I - y^H x)(I - x^H x)^{-1}(I - x^H y) - (y^H - x^H)(I - xx^H)^{-1}(y - x)$. Note that

$$(I - y^H x)(I - x^H x)^{-1}(I - x^H y) = (I - x^H x)^{-1} - (I - x^H x)^{-1}x^H y - y^H x(I - x^H x)^{-1} + y^H x(I - x^H x)^{-1}x^H y,$$

$$(y^H - x^H)(I - xx^H)^{-1}(y - x) = y^H(I - xx^H)^{-1}y - y^H(I - xx^H)^{-1}x - x^H(I - xx^H)^{-1}y + x^H(I - xx^H)^{-1}x.$$

Therefore

$$A = \left((I - x^H x)^{-1} - x^H(I - xx^H)^{-1}x\right) - \left((I - x^H x)^{-1}x^H - x^H(I - xx^H)^{-1}\right)y$$
$$- y^H\left(x(I - x^H x)^{-1} - (I - xx^H)^{-1}x\right) - y^H\left((I - xx^H)^{-1} - x(I - x^H x)^{-1}x^H\right)y.$$

By Lemma D.4,

$$(I - x^H x)^{-1} - x^H(I - xx^H)^{-1}x = I,$$
$$(I - x^H x)^{-1}x^H - x^H(I - xx^H)^{-1} = \mathbf{0}_n,$$
$$x(I - x^H x)^{-1} - (I - xx^H)^{-1}x = \mathbf{0}_n,$$
$$(I - xx^H)^{-1} - x(I - x^H x)^{-1}x^H = I.$$

Therefore

$$A = I - y^H y. \tag{17}$$

Combining Eqs. (16) and (17) leads to the conclusion of Lemma D.5.

$\square$

*Proof.* The distance $d_{\mathbb{SD}_n}(\cdot, \cdot)$ associated with the Kähler metric in the Siegel disk is given (up to scaling) by

$$d^2_{\mathbb{SD}_n}(x, y) = \text{Tr}\left(\log^2\left((I + c^{\frac{1}{2}})(I - c^{\frac{1}{2}})^{-1}\right)\right),$$

where $x, y \in \mathbb{SD}_n$ and $c$ is computed as

$$c = (y - x)(I - x^H y)^{-1}(y^H - x^H)(I - xy^H)^{-1}.$$

Since the distance $d_{\mathbb{SD}_n}(\cdot, \cdot)$ is invariant w.r.t. the automorphism $\phi_x(\cdot)$, we have

$$d^2_{\mathbb{SD}_n}(x, y) = d^2_{\mathbb{SD}_n}(\mathbf{0}_n, \phi_x(y))$$
$$= \text{Tr}\left(\log^2\left((I + \bar{c}^{\frac{1}{2}})(I - \bar{c}^{\frac{1}{2}})^{-1}\right)\right),$$

where $\bar{c}$ is computed as

$$\bar{c} = (\phi_x(y) - \mathbf{0}_n)\left(I - \mathbf{0}_n^H\phi_x(y)\right)^{-1}\left(\phi_x(y)^H - \mathbf{0}_n^H\right)\left(I - \mathbf{0}_n\phi_x(y)^H\right)^{-1}.$$

By Lemma D.5, we obtain the two formulae in Proposition 6.1.

$\square$

### D.6. Proof of Proposition 6.2

*Proof.* Given

$$z = (I - xx^H)^{-\frac{1}{2}}(y - x)(I - x^H y)^{-1}(I - x^H x)^{\frac{1}{2}}.$$

We need to prove that

$$y = (I - xx^H)^{\frac{1}{2}}(I + zx^H)^{-1}(z + x)(I - x^H x)^{-\frac{1}{2}}.$$

Let $\tilde{z} = (I - xx^H)^{\frac{1}{2}} z (I - x^H x)^{-\frac{1}{2}}$, then

$$\tilde{z} = (y - x)(I - x^H y)^{-1}.$$

Therefore

$$y = (I + \tilde{z} x^H)^{-1}(\tilde{z} + x).$$

Note that

$$
\begin{aligned}
(I + \tilde{z} x^H)^{-1} &= \left( I + (I - xx^H)^{\frac{1}{2}} z (I - x^H x)^{-\frac{1}{2}} x^H \right)^{-1} \\
&= \left( I + (I - xx^H)^{\frac{1}{2}} z x^H (I - xx^H)^{-\frac{1}{2}} \right)^{-1} \\
&= \left( (I - xx^H)^{\frac{1}{2}} (I + zx^H)(I - xx^H)^{-\frac{1}{2}} \right)^{-1} \\
&= (I - xx^H)^{\frac{1}{2}} (I + zx^H)^{-1}(I - xx^H)^{-\frac{1}{2}},
\end{aligned}
\tag{18}
$$

where the second equation follows from Lemma D.4.

Note also that

$$
\begin{aligned}
\tilde{z} + x &= (I - xx^H)^{\frac{1}{2}} z (I - x^H x)^{-\frac{1}{2}} + x \\
&= (I - xx^H)^{\frac{1}{2}} z (I - x^H x)^{-\frac{1}{2}} + (I - xx^H)^{\frac{1}{2}} x (I - x^H x)^{-\frac{1}{2}} \\
&= (I - xx^H)^{\frac{1}{2}} (z + x)(I - x^H x)^{-\frac{1}{2}},
\end{aligned}
\tag{19}
$$

where the second equation follows from Lemma D.4.

Combining Eqs. (18) and (19), one has

$$y = (I - xx^H)^{\frac{1}{2}}(I + zx^H)^{-1}(z + x)(I - x^H x)^{-\frac{1}{2}}.$$

Since $y = y^T$, $z = z^T$, and $x = x^T$, one has

$$
\begin{aligned}
y &= (I - xx^H)^{-\frac{1}{2}}(z + x)(I + x^H z)^{-1}(I - x^H x)^{\frac{1}{2}} \\
&= \phi_{-x}(z).
\end{aligned}
$$

This completes the proof of the proposition.

$\square$

### D.7. Proof of Corollary 6.3

*Proof.* The automorphism $\phi_x(\cdot)$ in Eq. (2) satisfies the property that $\phi_{0_{\mathbb{D}}}(tx) = tx = t\phi_{0_{\mathbb{D}}}(x)$ for any $x \in \mathbb{SD}_n$ and $t \in [0, 1]$. Thus Corollary 6.3 is a consequence of Proposition 5.2 when $q(x) = \|x\|_2$.

$\square$

### D.8. Proof of Corollary 6.4

*Proof.* The automorphism $\phi_x(\cdot)$ in Eq. (5) satisfies the property that $\phi_{0_{\mathbb{D}}}(tx) = tx = t\phi_{0_{\mathbb{D}}}(x)$ for any $x \in \mathbb{B}_n$ and $t \in [0, 1]$. Thus Corollary 6.4 is a consequence of Proposition 5.2 when $q(x) = |x|$.

$\square$

