# OpenReview forum: "Batch Normalization for Neural Networks on Complex Domains"
_ICML.cc/2026/Conference — ICML 2026 regular_

### Official Review · Reviewer_WspG · 2026-03-12

**Soundness:** 3
**Presentation:** 3
**Significance:** 3
**Originality:** 3
**Overall Recommendation:** 4
**Confidence:** 4

**Summary:**

This study proposes a new Batch Normalization (BN) method for neural networks working in complex spaces, specifically focusing on the Siegel disk and the complex unit ball. Instead of using the usual manifold exponential and logarithmic maps, the authors use automorphisms to center and adjust the batches.

**Compliance With Llm Reviewing Policy:**

Affirmed.

**Final Justification:**

Rebuttal has addressed some of the experimental verification issues. I'd like to keep my original score.

**Key Questions For Authors:**

1	Why was variance scaling explicitly omitted from the formulation? Is it theoretically intractable to define a coherent scaling operation alongside the Kobayashi pseudodistance, or was it purely an empirical choice?
2	Your method computes the Fréchet mean via Euclidean parameterization and standard gradient descent (Eq. 10). Can you elaborate on the practical convergence stability? Specifically, does the lack of a closed-form Fréchet mean computation introduce instability, vanishing, or exploding gradients during the network's backward pass?
3	The radar clutter datasets utilized in your primary task are entirely synthetic. Have you tested this normalization technique on any real-world radar or complex signal datasets to verify its robustness to non-ideal noise distributions?

**Limitations:**

While the authors transparently acknowledge the computational bottlenecks associated with SVD operations and Fréchet mean computation in Appendix C, they do not adequately discuss the theoretical and practical limitations of omitting variance scaling, nor do they address the potential lack of external validity due to relying on purely synthetic data for their primary task.

**Strengths And Weaknesses:**

Strengths:
-Extending BN layers to complex domains using automorphisms and almost geodesics is a highly original and elegant contribution to geometric deep learning, effectively addressing domains where closed-form maps are unavailable.
-The mathematical foundation is rigorous, establishing clear, proven connections and reductions to existing Riemannian BN variants across gyrogroups, Lie groups, and symmetric spaces
Weaknesses:
-The proposed BN layer purely performs centering and biasing, explicitly omitting variance scaling. Standard Euclidean and Riemannian BNs rely heavily on variance scaling to stabilize the training dynamics. Calling this "Batch Normalization" may cause confuse, and the lack of theoretical or empirical discussion regarding how the absence of scaling affects manifold dispersion is a significant gap.
-In the node classification task (CBallNetBN), the pipeline utilizes Discrete Fourier Transforms (DFT) to map features back and forth between the Poincaré ball and the complex unit ball. The current ablation studies do not clearly isolate whether the performance gains originate inherently from the BN layer, or from the interaction of the data with the DFT domain conversions
-Computing the batch mean requires iteratively solving for the Fréchet mean using gradient descent parameterized in Euclidean space. Combined with the reliance on heavy matrix operations for Siegel spaces, the forward and backward passes are computationally expensive, severely limiting scalability for large-scale or high-dimensional problems.

---

> ### Author Rebuttal · Authors · 2026-03-31
>
> We thank the reviewer for their insightful comments on the paper. We appreciate the questions and suggestions for improving it. Please find below our responses to the reviewer questions.
>
> **Q1: Why was variance scaling explicitly omitted from the formulation?**
>
> Variance scaling can be introduced in our BN layer using the following transformation
> \begin{equation}
> \tilde{x}\_j = \phi_g^{(-1)} \left ( \frac{s}{\sqrt{v + \epsilon}} \otimes \phi\_{m\_b}(x\_j) \right ),
> \end{equation}
> where $v$ is the empirical variance of the batch of $k$ points $\\{ x\_j \\}_{j=1}^k$ (see the notations in Algorithm 1), $s$ is a scale parameter, $\otimes$ is the scalar multiplication in Definition 5.1, and $\epsilon$ is an arbitrarily small positive constant for numerical stability. Table B reports the results of SiegelNetBN and SiegelNetScaleBN which uses the above normalization method.
>
> |Method|D1|D2|D3|D4|D5|D6|
> |--|--|--|--|--|--|--|
> |SiegelNetBN|33.83$\pm$0.34|43.29$\pm$0.40|41.38$\pm$0.37|**65.93$\pm$0.32**|**45.61$\pm$0.12**|37.10$\pm$0.26|
> |SiegelNetScaleBN|**38.42$\pm$0.41**|**47.32$\pm$0.79**|**43.23$\pm$0.12**|65.46$\pm$0.86|45.20$\pm$0.04|**37.46$\pm$0.21**|
>
> Table B. Impact of variance scaling on our BN layer.
>
> Table B shows that variance scaling is beneficial for our BN and yields substantial performance gains on some datasets. However, from a theoretical perspective, it is important to provide a statistical analysis on the impact of the scaling operation in our framework similar to the one discussed in Chakraborty (2020) (see Proposition 3). There is also the practical question of whether one can design a better scalar multiplication and an effective variance scaling scheme [A8, A9]. Finding the answers to these questions requires further investigation. Although the scaling operation is absent in our formulation, we adopt the approach in Brooks et al. (2019) which relies solely on the centering and biasing operations and thus name our method as BN on complex domains.
>
> **Q2: Does the lack of a closed-form Fréchet mean computation introduce instability, vanishing, or exploding gradients during the network's backward pass?**
>
> Our networks are not prone to the vanishing or exploding gradient issues due to their shallow architecture. However, in our implementation, parameter initialization is carefully done to make sure that these issues do not happen. A key challenge of our Siegel network is that it relies on SVD operations which might lead to numerical instability.  This problem is alleviated using the following methods [A7]: (1) enable double precision floating-point format in computations; (2) use SVD to compute eigenvalues and eigenvectors; and (3) clamp eigenvalues to make them positive (see Appendix B.1.2). In our experiments, we did not encounter any of the mentioned issues during network training.
>
> **Q3: Have you tested this normalization technique on any real-world radar or complex signal datasets to verify its robustness to non-ideal noise distributions?**
>
> We agree that testing on such datasets would strengthen the paper which we could not.
> We used synthetic data due to the scarcity of publicly available datasets of real-world radar signals. Generating synthetic radar signals using the simulation model in [A10; Cabanes & Nielsen, 2021] also ensures that the signals are stationary centered autoregressive Gaussian time series. This assumption is required in order for Algorithm 2 (see Appendix B.1.2) to output data belonging to products of Hermitian positive definite space and Siegel spaces. Therefore, the performance of our method is best demonstrated when the assumption holds. In practice, however, the assumption is not strictly required because we can project the output of Algorithm 2 to Siegel spaces before feeding them to our network. The experiments in Appendix B.3 show that our method remains effective even in some settings where the assumption does not hold. Also, noise distributions are not ideal in these settings. This demonstrates that our method can potentially be used for analyzing real-world radar and complex signals.
>
> **References**
>
> [A8] Aaron Lou, Isay Katsman, Qingxuan Jiang, Serge J. Belongie, Ser-Nam Lim, Christopher De Sa: Differentiating through the Fréchet Mean. ICML 2020: 6393-6403.
>
> [A9] Johan Bjorck, Carla P. Gomes, Bart Selman, Kilian Q. Weinberger: Understanding Batch Normalization. NeurIPS 2018: 7705-7716.
>
> [A10] Yann Cabanes, Frédéric Barbaresco, Marc Arnaudon, Jérémie Bigot: Toeplitz Hermitian Positive Definite Matrix Machine Learning Based on Fisher Metric. GSI 2019: 261-270.

---

> > ### Author Rebuttal · Reviewer_WspG · 2026-04-04
> >
> > Experimental verification has been added. I'd like to keep my positive score.

---

> > > ### Author Response · Authors · 2026-04-06
> > >
> > > We would like to thank the reviewer for your careful consideration of our rebuttal and your positive evaluation of our work.

---

### Official Review · Reviewer_5KxN · 2026-03-12

**Soundness:** 3
**Presentation:** 3
**Significance:** 3
**Originality:** 3
**Overall Recommendation:** 4
**Confidence:** 4

**Summary:**

This paper focuses on extending batch normalization techniques to neural networks defined on complex domains within the Riemannian learning framework. Building upon prior work on Riemannian batch normalization, the authors derive the theoretical components necessary for implementing BN layers on complex manifolds that have received limited attention in previous studies, such as the Siegel disk domain. The proposed approach aims to improve training stability and representation learning in Riemannian neural networks. Experimental evaluations on radar clutter classification, node classification, and action recognition tasks demonstrate the effectiveness of the method.

**Compliance With Llm Reviewing Policy:**

Affirmed.

**Key Questions For Authors:**

1.How does the proposed BN compare with other normalization methods such as Layer Normalization, Group Normalization, or Weight Normalization when applied to networks operating on complex domains?

2.What is the computational cost of the proposed BN layer compared with standard BN and existing Riemannian BN approaches？

3.This paper focuses on complex domains such as the Siegel disk. Could the proposed framework be easily extended to other Riemannian manifolds beyond those discussed in the paper?

4.Can the proposed approach scale to large-scale architectures or datasets, such as deep CNNs or transformer-based models operating in geometric spaces?

**Limitations:**

Yes

**Strengths And Weaknesses:**

Strengths：

1.This paper investigates the design of normalization layers for neural networks operating in complex domains and on less explored Riemannian manifolds. The proposed study broadens the applicability of normalization techniques in geometric deep learning, whereas existing research has primarily focused on manifolds such as SPD manifolds or hyperbolic spaces.

2.This paper derives the formulation of batch normalization (BN) layers from a Riemannian geometric perspective, providing a relatively rigorous mathematical framework for implementing batch normalization in complex domains.

3.The experiments cover several tasks across different domains (e.g., radar clutter classification, node classification, and action recognition), which helps demonstrate the general applicability of the proposed approach.

Weaknesses：

1.The derivations related to the geometric structure of complex domains and the Siegel disk are relatively involved. For readers who are not familiar with Riemannian geometry, these parts may be difficult to follow. Providing more intuitive explanations or illustrative examples could help improve the overall readability of the paper.

2.Although the paper discusses practical implementation aspects, it lacks a detailed analysis of the computational overhead and complexity.

3.Although several tasks are included, the experiments are relatively limited in scale. It would strengthen this method to evaluate the proposed BN layer on larger benchmarks or modern architectures.

---

> ### Author Rebuttal · Authors · 2026-03-30
>
> We thank the reviewer for their insightful comments on the paper. We appreciate the questions and suggestions for improving it. Please find below our responses to the reviewer questions.
>
> **Q1: How does the proposed BN compare with other normalization methods such as Layer Normalization, Group Normalization, or Weight Normalization?**
>
> It would be interesting to compare our BN with these normalization methods within the same network architecture such as SiegelNetBN. However, building these layers on Siegel spaces is beyond the scope of our paper. Therefore, we shall focus on comparing SiegelNetBN and ComplexLSTM (see Table 1) with complex layer normalization (CLN), an extension of
> (real) layer normalization [A1] for complex values. We implement CLN using Eqs. (17), (18), and (19) in [A1].
> The method in [A2] is used to normalize an array of complex numbers. Results in Table A show that ComplexLSTM-CLN generally outperforms ComplexLSTM, but the former is inferior to our network.
>
> |Method|D1|D2|D3|D4|D5|D6|
> |----|----|----|----|------|-----|-----|
> |ComplexLSTM-CLN |26.62$\pm$0.39|26.53$\pm$0.85|27.82$\pm$0.39|47.82$\pm$3.53|19.75$\pm$0.90|24.92$\pm$0.46|
> | SiegelNetBN (Ours) |**33.83$\pm$0.34**|**43.29$\pm$0.40**|**41.38$\pm$0.37**|**65.93$\pm$0.32**|**45.61$\pm$0.12**|**37.10$\pm$0.26**|
>
> Table A: Comparison of our BN and layer normalization.
>
> **Q2: What is the computational cost of the proposed BN layer?**
>
> We analyze the computational costs in the training stages of our BN layer and the well-established BN layer (SPDBN) in (Brooks et al., 2019).  Denote by $n\_{iters}$ the number of iterations for Fréchet mean computation, $k$ the batch size, $n$ the dimension of input matrices. The steps of our BN layer have the following time complexities:
> - Fréchet mean computation: $O(n\_{iters}kn^3)$
> - Running mean update: $O(n^3)$
> - Centering and biasing points: $O(kn^3)$
>
> Overall, our BN layer has a time complexity of order $O(n\_{iters}kn^3)$ which is also the time complexity of the SPDBN layer. Here we assume that $n\_{iters}$ is the number of iterations for Karcher mean computation in the SPDBN layer.
>
> **Q3: Could the proposed framework be easily extended to other Riemannian manifolds?**
>
> The Siegel disk is holomorphically isomorphic to the Siegel upper half-space which belongs to Siegel domains [A3]. It has been shown that the automorphisms of a large family of Siegel domains can be characterized and that their automorphism groups  have Lie group structures. Therefore, the proofs of most of our theoretical results can be adapted to a more general setting, and extensions of our framework can potentially be used for Siegel domains in [A4] (page 149), [A5] (page 318), and [A6] (page 3).
>
> **Q4: Can the proposed approach scale to large-scale architectures or datasets?**
>
> Basic building blocks in deep CNNs, e.g., fully-connected and convolutional layers can be extended to Siegel spaces using our approach and an idea similar to Shimizu et al. (2021). These are also key components in transformer-based models. Since there do not exist closed forms for the Fréchet mean and geometric quantities (the exponential map, logarithmic map, and parallel transport) on Siegel spaces, a key challenge is how to efficiently compute the Fréchet mean in order to realize attention mechanisms. The same applies to learning with graph convolutional networks (GCNs) in which the feature aggregation operation of a basic GCN message-passing update relies on the Fréchet mean. Note that this is also a challenge for many deep models operating in geometric spaces. For instance, feature aggregation in SPD and Grassmann GCNs [A7; Nguyen et al., 2024] and attention mechanisms in SPD networks [Pan et al., 2022] are performed on tangent spaces to avoid cumbersome Riemannian computations. Please refer to Appendix B.3 for our experiments on large-scale datasets.
>
> **References**
>
> [A1] J. L. Ba, J. R. Kiros, and G. E. Hinton, Layer normalization, arXiv preprint arXiv:1607.06450 (2016).
>
> [A2] C. Trabelsi, O. Bilaniuk, Y. Zhang, D. Serdyuk, S. Subramanian, J. F. Santos, S. Mehri, N. Rostamzadeh, Y. Bengio, and C. J. Pal. Deep complex networks. In ICLR, 2018.
>
> [A3] I. I. Pjateckii-Sapiro, Géométrie des domaines classiques et théorie des fonctions automorphes, Dunod, Paris, 1966.
>
> [A4] É. Cartan, Sur les domaines bornés homogènes de l'espace de n variables complexes, Abh. Math. Sem. Univ. Hamburg 11 (1935), 116-162.
>
> [A5] B.A. Fuks, Special Chapters in the Theory of Analytic Functions of Several Complex Variables, Transl. of Math. Monographs 14, Amer. Math, Soc., Providence, R.I., 1965.
>
> [A6] M. Morimoto, Analytic functionals on the Lie sphere, Tokyo J. Math. 3 (1980), 1–35.
>
> [A7] Zhao, W., Lopez, F., Riestenberg, M.J., Strube, M., Taha, D., Trettel, S.: Modeling graphs beyond hyperbolic: Graph neural networks in symmetric positive definite matrices. In: Joint European Conference on Machine Learning and Knowledge Discovery in Databases, pp. 122-139 (2023).

---

> > ### Author Rebuttal · Reviewer_5KxN · 2026-04-03
> >
> > Thanks for your response, which addresses my concerns. I will keep my score.

---

> > > ### Author Response · Authors · 2026-04-06
> > >
> > > We would like to thank the reviewer for your feedback. We are glad that your concerns have been fully resolved. The analysis and experiments provided in our rebuttal will be added to the Appendix on reviewer demand.

---

### Official Review · Reviewer_K7Qw · 2026-03-12

**Soundness:** 3
**Presentation:** 2
**Significance:** 3
**Originality:** 3
**Overall Recommendation:** 4
**Confidence:** 5

**Summary:**

This paper proposes Batch Normalization (BN) layers for neural networks defined on complex domains. The proposed formulation is closely related to existing Riemannian BN methods. The authors derive components needed for the practical implementation of BN layers on several complex domains that have received limited attention in prior work, including the Siegel disk domain.  Evaluations are performed using radar clutter classification, node classification, and action recognition, and improved results are reported.

**Compliance With Llm Reviewing Policy:**

Affirmed.

**Key Questions For Authors:**

Although the problem addressed in this paper is important, the motivation of this paper is weakly stated. The question of why we need to develop another BN is unanswered.
The results provide some support for the statement of improvements, but it is unclear if the proposed BN can be integrated into various networks and achieve reasonable performance gains.
Methods such as ps Chen et al., 2024 and Chen et al., 2025 should be considered in the experiments.

**Limitations:**

yes

**Strengths And Weaknesses:**

This paper studies a component in deep networks. This component often serves as a key factor for real-world applications to achieve learnability with diverse datasets.
Although the problem addressed in this paper is important, the motivation of this paper is weakly stated. The question of why we need to develop another BN is unanswered.
The results provide some support for the statement of improvements, but it is unclear if the proposed BN can be integrated into various networks and achieve reasonable performance gains.
Methods such as ps Chen et al., 2024 and Chen et al., 2025 should be considered in the experiments.

---

> ### Author Rebuttal · Authors · 2026-03-30
>
> We thank the reviewer for their insightful comments on the paper. We appreciate the questions that allow us to clarify our ideas.
> Please find below our responses to the reviewer questions.
>
> **Q1: The motivation of this paper is weakly stated. The question of why we need to develop another BN is unanswered.**
>
> Many complex domains (see Section 5) have the potential to capture rich geometrical structures for representation learning but they have attracted little attention. An example of such domains is Siegel spaces which generalize hyperbolic and SPD manifolds, two widely used representation spaces in Riemannian neural networks. These domains have nice theoretical properties, e.g., the characterizations of automorphisms can be given, and the groups of automorphisms have Lie group structures. These properties make them an attractive tool for building neural networks on non-Euclidean spaces. While a number of Riemannian BN layers have been proposed, their counterpart operating in these domains seems to be missing. This motives us to propose a BN layer which is mainly designed for these domains, but can be seamlessly adapted to the Riemannian and Lie group settings commonly encountered in previous works.
>
> **Q2: It is unclear if the proposed BN can be integrated into various networks and achieve reasonable performance gains.**
>
> As shown in the paper, our experimental results on two complex domains and three machine learning tasks have demonstrated the applicability of the proposed method. In particular, our networks consistently outperform their competitors on all the datasets. It would be interesting to investigate the effectiveness of our BN layer in complex domains that are not considered in our work (please refer to our answer to question 3 of Reviewer 5KxN). In the case of SPD manifolds, our method based on the automorphism and scalar multiplication defined in Section 4.1 will produce similar results as the state-of-the-art network SPDNetBN in Brooks et al. (2019).
>
> **Q3: Methods such as ps Chen et al., 2024 and Chen et al., 2025 should be considered in the experiments.**
>
> We did not consider these BN methods in the radar clutter classification experiments because they were originally proposed for Lie groups and (pseudo-reductive) gyrogroups which are not applicable to the studied complex domains. Please refer to Appendix B.3 for a comparison of our method and these methods on the human action recognition task.

---

> > ### Author Rebuttal · Reviewer_K7Qw · 2026-04-03
> >
> > I believe the authors clarified a few points, but the major weakness still exists.

---

> > > ### Author Response · Authors · 2026-04-06
> > >
> > > We would like to thank the reviewer for your prompt feedback and your careful consideration of our rebuttal.

---

### Decision · Program_Chairs · 2026-04-30

**Decision:**

Accept (regular)

**Comment:**

This paper proposes a batch-normalization layer for neural networks on complex domains, with concrete instances on the Siegel disk and the complex unit ball. The main idea is to perform centering, biasing, and running-mean updates using automorphisms and almost geodesics, avoiding reliance on closed-form exponential and logarithmic maps. The paper also relates the construction to existing Riemannian BN methods and provides experiments on several tasks.

All reviewers gave weak accept recommendations. Their concerns about motivation, evaluation breadth, and scalability are reasonable, but I do not view them as decisive, and the rebuttal addressed them reasonably well. I am inclined to accept this paper because it presents a novel and mathematically coherent viewpoint for building normalization layers on geometric domains where standard Riemannian tools are not easily available. While somewhat specialized, this perspective could be interesting to researchers in geometric deep learning and may help broaden the range of spaces where principled neural-network layers can be developed.